# Non-canonical H3K79me2-dependent pathways promote the survival of MLL-rearranged leukemia

William F Richter[1], Rohan N Shah[1,2], Alexander J Ruthenburg[1,3]*

[1]Department of Molecular Genetics and Cell Biology, The University of Chicago, Chicago, United States; [2]Pritzker School of Medicine, The University of Chicago, Chicago, United States; [3]Department of Biochemistry and Molecular Biology, The University of Chicago, Chicago, United States

**Abstract** MLL-rearranged leukemia depends on H3K79 methylation. Depletion of this transcriptionally activating mark by DOT1L deletion or high concentrations of the inhibitor pinometostat downregulates *HOXA9* and *MEIS1*, and consequently reduces leukemia survival. Yet, some MLL-rearranged leukemias are inexplicably susceptible to low-dose pinometostat, far below concentrations that downregulate this canonical proliferation pathway. In this context, we define alternative proliferation pathways that more directly derive from H3K79me2 loss. By ICeChIP-seq, H3K79me2 is markedly depleted at pinometostat-downregulated and MLL-fusion targets, with paradoxical increases of H3K4me3 and loss of H3K27me3. Although downregulation of polycomb components accounts for some of the proliferation defect, transcriptional downregulation of FLT3 is the major pathway. Loss-of-FLT3-function recapitulates the cytotoxicity and gene expression consequences of low-dose pinometostat, whereas overexpression of constitutively active *STAT5A*, a target of FLT3-ITD-signaling, largely rescues these defects. This pathway also depends on MLL1, indicating combinations of DOT1L, MLL1 and FLT3 inhibitors should be explored for treating *FLT3*-mutant leukemia.

**\*For correspondence:**
aruthenburg@uchicago.edu

**Competing interests:** The authors declare that no competing interests exist.

## Introduction

MLL1-rearrangements (MLL-r) account for ~10% of all leukemia cases and are especially prominent in infants (70–80%) and, lacking an effective standard of care, bear a very poor prognosis (*Marks et al., 2013*; *Jabbour et al., 2015*; *Mann et al., 2010*; *Pieters et al., 2007*; *Winters and Bernt, 2017*). A growing body of evidence suggests that MLL-rearrangements rely on additional mutations to cause leukemia. Leukemia patients with MLL-fusions often have additional mutations that affect growth signaling pathways (*Grossmann et al., 2013*; *Liang et al., 2006*; *Armstrong et al., 2003*) and MLL-fusions in mouse models cause leukemias with longer-than-expected latencies, suggesting that additional mutations are required for full progression (*Ono et al., 2005*; *Corral et al., 1996*; *Forster et al., 2003*). Yet, few studies have examined the genetic context of MLL-fusion proteins and how additional lesions may cooperate to promote disease at the molecular level.

*MLL1* (Mixed Lineage Leukemia protein, also known as *KMT2A*) is a histone H3 lysine methyltransferase involved in regulating *HOX* gene expression during development and normal hematopoiesis (*Hess, 2004*). Translocations of MLL1 fuse its amino terminus to the carboxy-terminus of a growing list of over 130 different fusion partners (*Meyer et al., 2018*). Although these MLL-fusions lack methyltransferase activity, a functional copy of the MLL1 gene is necessary to target and hypermethylate H3K4 at MLL-fusion target genes to induce leukemogenesis (*Milne et al., 2005*; *Cao et al., 2014*; *Milne et al., 2010*). In more than 75% of acute myeloid leukemia (AML) cases and >90% of acute

lymphoblastic leukemia (ALL) cases involving MLL translocations, the MLL-fusion partner is one of seven members of the transcriptional elongation complex, most commonly, AF9 and AF4, respectively (*Marschalek, 2011*). These fusion partners aberrantly recruit DOT1L, the sole histone H3 lysine 79 methyltransferase to MLL1 target genes including the HOXA gene cluster (*Mohan et al., 2010*; *Okada et al., 2005*; *Kerry et al., 2017*). By mechanisms that remain unclear, DOT1L-mediated hypermethylation of H3K79 promotes expression of MLL-fusion targets (*Milne et al., 2005*; *Bernt et al., 2011*; *Guenther et al., 2008*; *Stubbs et al., 2008*; *Chen et al., 2015a*), establishing an expression profile with a surprising degree of target gene overlap across different MLL-fusions (*Armstrong et al., 2002*). Ablation of H3K79 methylation through knockout or pharmacological targeting of *DOT1L* abrogates the MLL-fusion target gene expression profile, selectively induces apoptosis and differentiation of leukemia cells in culture and dramatically extends the survival of mice in xenograft experiments (*Bernt et al., 2011*; *Daigle et al., 2013*).

Viral co-transduction of the MLL-AF4 targets (*Zeisig et al., 2004*) H*OXA9* and *MEIS1* is sufficient to cause acute leukemia in mouse bone marrow progenitors, arguing that these transcription factors represent a major etiologic pathway in MLL-r leukemia (*Corral et al., 1996*; *Chang et al., 2010a*; *Jo et al., 2011*; *Kroon et al., 1998*; *Calvo et al., 2002*). However, exogenous expression of *MLL-AF9* in mice requires a long latency period (4–9 months) and chemotherapy induced MLL-translocations cause disease 3–5 years after treatment, suggesting that additional mutations are required for leukemagenesis (*Corral et al., 1996*; *Dobson et al., 1999*). In the prevailing model, MLL-fusions recruit DOT1L to hypermethylate and activate expression of *MEIS1* and *HOXA9* (*Figure 1A*; *Okada et al., 2005*; *Bernt et al., 2011*; *Guenther et al., 2008*; *Daigle et al., 2011*; *Deshpande et al., 2013*). However, the genetic manipulations used to define this paradigm may have missed more subtle and graded effects afforded by kinetically-staged antagonism with highly specific small-molecule inhibitors. Therefore, to better understand the direct effects of H3K79me2 in several MLL-r cell lines we employed pharmacologic inhibition of DOT1L methyltransferase activity.

Pinometostat (EPZ5676), a highly specific DOT1L inhibitor (*Daigle et al., 2013*; *Daigle et al., 2011*; *Anglin and Song, 2013*; *Yu et al., 2012*) displays 37,000-fold selectivity over its closest related paralogs and a host of other lysine and arginine methyltransferases. Interestingly, several cell lines that all have the *MLL-AF4* translocation display pinometostat sensitivities that differ by nearly three orders of magnitude (*Daigle et al., 2013*). One of these lines (MV4;11) displays a pinometostat IC50 for proliferation that is 20 times lower than the IC50 for *HOXA9* and *MEIS1* expression (*Daigle et al., 2013*), suggesting that these drivers of leukemogenesis, though downregulated at higher concentrations (1 µM) (*Daigle et al., 2013*), may not contribute to cell-type-specific effects at lower concentrations.

We sought to understand low-dose pinometostat effects by treating a variety of MLL-r cell lines with a concentration that reduces proliferation in only a subset, with MLL-r cell lines harboring *FLT3-ITD* mutations being the most susceptible. Under these conditions, *HOXA9* and *MEIS1* expression remain unaffected, presenting a clear exception to the existing paradigm, but we found thousands of other differentially expressed genes, including the *PBX3* and *FLT3* oncogenes. Capitalizing on the sensitivity of internally calibrated ChIP-seq (ICeChIP-seq) (*Grzybowski et al., 2015*; *Grzybowski et al., 2019*), we observed larger reductions in H3K79me2 density at a subset of MLL-AF4 targets, a genome-wide reduction in H3K27me3 and stark H3K4me3 increases at transcription start sites. Remarkably, we could nearly completely rescue not only pinometostat- but also MLL1 inhibitor-induced effects on proliferation and apoptosis through expression of a constitutively active form of the downstream FLT3-ITD target *STAT5A* (*STAT5A-CA*), arguing that disruptions to this pathway represent the main source of toxicity from low-dose DOT1L inhibition. In addition, DOT1L inhibition also downregulated the *EZH2* and *EED* components of the PRC2 complex, likely accounting for global reductions in H3K27me3 and imparting modest, but distinct effects on proliferation and a correspondingly moderate proliferation rescue from EZH2 overexpression. Collectively, our data argue that the FLT3-ITD signaling and PRC2 pathways, are more sensitive to disruptions of MLL-fusion-mediated gene activation than the canonical oncogenic drivers in MLL-r, *FLT3*[ITD] leukemias, defining a new molecular understanding of how MLL-fusions cooperate with other oncogenic factors to induce leukemia.

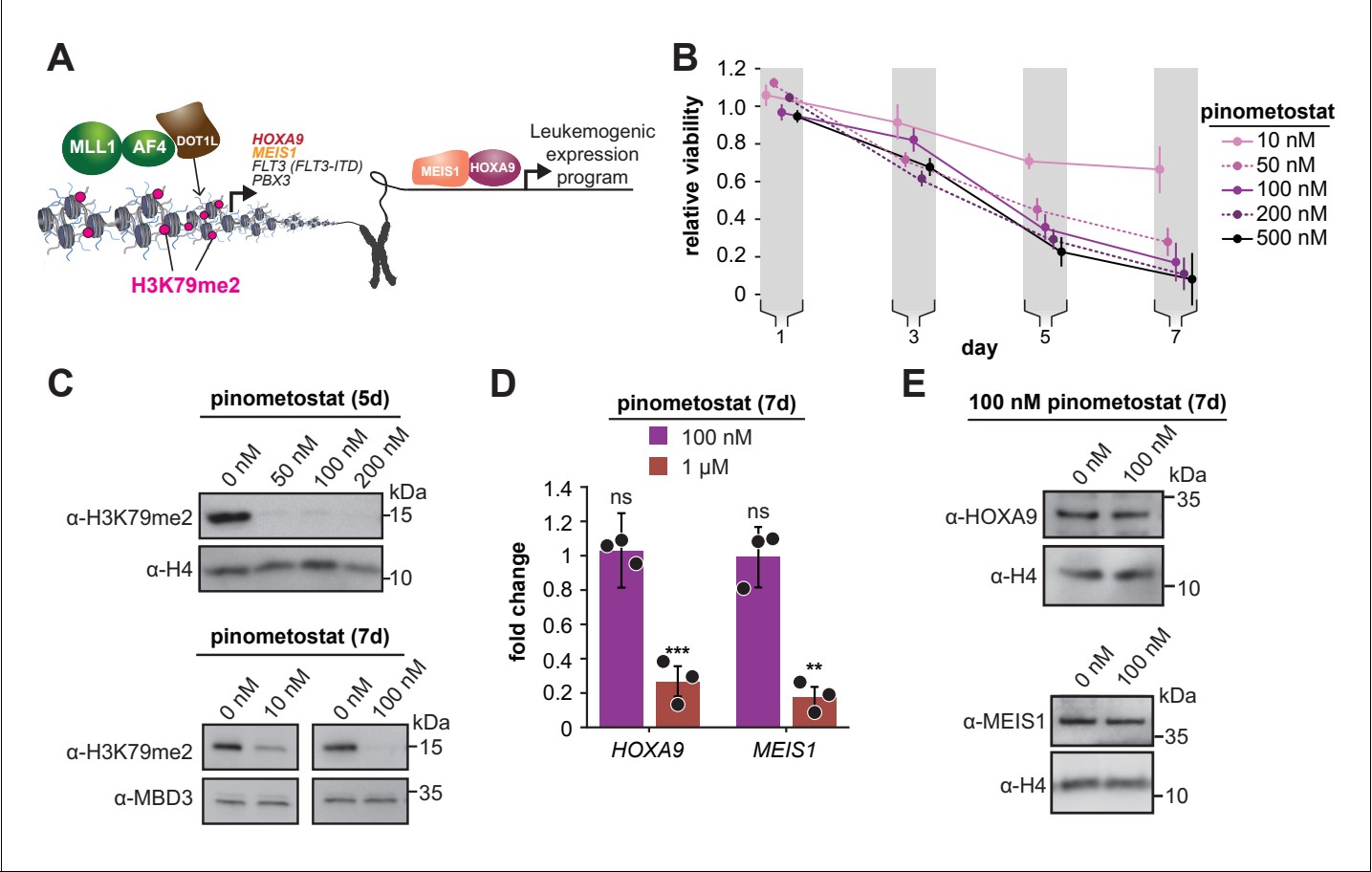

**Figure 1.** Low doses of DOT1L inhibitor ablate bulk H3K79me2 and curtail MV4;11 proliferation without impacting expression of canonical target genes. (**A**) Conventional model depicting how DOT1L methyltransferase activity activates transcription of key proliferative oncogenic transcription factors (*Okada et al., 2005*; *Bernt et al., 2011*; *Guenther et al., 2008*; *Armstrong et al., 2002*; *Zeisig et al., 2004*; *Kroon et al., 1998*). (**B**) Proliferation assay of MV4;11 cells treated with the indicated concentrations of the DOT1L inhibitor pinometostat (EPZ5676). Cell viability was assayed every 2 days, starting 1 day after treatment commenced using the CellTiter Glo 2.0 reagent. Relative cell viability is presented as the mean fraction of pinometostat versus cells treated with the equivalent volume of DMSO from three independent experiments ± S.E.M. (**C**) Western blots for H3K79me2 with H4 or MBD3 loading controls in MV4;11 cells treated with 10–200 nM pinometostat for 5 or 7 days. (**D**) RT-qPCR analysis of *HOXA9* and *MEIS1* expression fold-change in MV4;11 cells treated with 100 nM or 1 μM pinometostat for 7 days. Results are shown as mean ± S.E.M. of three independent experiments. Student's t-test (ns $p > 0.05$, ** $p \leq 0.01$, *** $p \leq 0.001$). (**E**) Western blot of HOXA9 and MEIS1 with H4 as a loading control from MV4;11 cells treated with 100 nM pinometostat for 7 days.

The online version of this article includes the following figure supplement(s) for figure 1:

**Figure supplement 1.** Low-dose DOT1L inhibition has little effect on Hox gene expression.

## Results

### MLL-r leukemia is sensitive to DOT1L inhibitor via a non-canonical pathway

Leukemias harboring MLL-rearrangements are uniquely susceptible to DOT1L inhibition and MV4;11, a biphenotypic leukemia cell line harboring an *MLL-AF4* translocation, is one of the most sensitive (*Daigle et al., 2013*). To determine the basis of this susceptibility, we systematically examined how low-dose regimes of pinometostat affect proliferation and global H3K79me2 levels in cells treated for 7 days with 10–500 nM pinometostat. This range of concentrations is slightly above the previously determined MV4;11 proliferation IC50 (3.5 nM) but is below the 1 μM or higher typically used in published investigations of the effects of H3K79me ablation (*Daigle et al., 2013*; *Godfrey et al., 2019*; *Okuda et al., 2017*). Consonant with previous findings (*Daigle et al., 2013*), pinometostat concentrations as low as 10 nM significantly reduce global levels of H3K79me2 and

cause a 30 ± 10% reduction in MV4;11 proliferation, while 100 nM inhibitor reduced cell proliferation by 80 ± 10% (*Figure 1B and C*). Notably, after treating MV4;11 cells with 100 nM inhibitor for 7 days we observed no discernable effect on the expression of *HOXA9* and *MEIS1* (*Figure 1D and E*), despite the emphasis on these genes as the critical mediators of DOT1L's effects in MLL-r leukemia (*Okada et al., 2005*; *Bernt et al., 2011*; *Guenther et al., 2008*; *Daigle et al., 2011*; *Deshpande et al., 2013*). Treatment with a low-dose regime of SGC0946, a distinct, yet highly selective DOT1L inhibitor (*Yu et al., 2012*) also reduced MV4;11 proliferation without affecting HOXA9 and MEIS1 expression (*Figure 1—figure supplement 1A and B*). Consistent with prior observations (*Daigle et al., 2013*), a much higher dose of 1 μM pinometostat significantly downregulates both *HOXA9* and *MEIS1* expression (*Figure 1D*).

## DOT1L inhibition at low concentrations downregulates leukemic oncogenes

With the extant model (*Okada et al., 2005*; *Bernt et al., 2011*; *Guenther et al., 2008*; *Daigle et al., 2011*; *Deshpande et al., 2013*) unable to explain reductions in proliferation caused by the DOT1L inhibitor in this concentration regime, we reasoned that the expression of other genes crucial to the survival of these cells are likely affected. To define these genes, we performed RNA-seq in MV4;11 cells that had been treated with 100 nM pinometostat for 7 days and observed that 1916 genes were downregulated and 2007 genes were upregulated (*Figure 2A*) relative to a DMSO treated control. To account for any handling biases, we included four RNA 'spike-in' controls and found no significant differences in read counts between treatment groups (*Figure 1—figure supplement 1C*). The downregulated genes significantly overlap with MLL-AF4 targets identified by Kerry et al. by ChIP-seq in MV4;11 cells (*Kerry et al., 2017*; *Figure 2B*). Relative to prior high-dose (3 μM) treatment with a compound structurally related to pinometostat in MV4;11 cells, the numbers of differentially expressed genes are similar, and there is marked overlap between the sets, particularly the downregulated cohort (*Daigle et al., 2011*; *Figure 2C* and *Figure 1—figure supplement 1D*). Consistent with our RT-qPCR measurements, *HOXA9* was unaltered in its expression (*Figure 1—figure supplement 1E*) and *MEIS1* displayed extremely modest mRNA reduction (20%) not observed by RT-qPCR and not reflected in apparent protein levels (*Figure 1D–E*). Of the other *HOXA* cluster genes only *HOXA11* and *HOXA13* exhibited expression changes with a 1.7-fold decrease and 2.5-fold increase, respectively (*Figure 1—figure supplement 1E*).

Although H3K79me2 is considered transcriptionally activating, the upregulated genes had much larger expression fold-changes. 906 genes were upregulated at least twofold (and some > 80-fold), while only 86 genes were downregulated ≥ 2-fold (*Figure 2A*). The list of upregulated transcripts includes MHC class II and innate immune response genes (*Figure 2D*). We confirmed the expression increases of *CIITA* (the master regulator of interferon-inducible MHC class II genes), and the MHC class II genes *HLA-DRA* and *HLA-DRB1* by RT-qPCR (*Figure 2E*). Gene ontology analysis of the upregulated genes indicated enrichment for 'immune response' and 'interferon-gamma signaling pathway' (*Figure 2—figure supplement 1A*; *Huang et al., 2009a*; *Huang et al., 2009b*). Despite there being no discernable effect on interferon-gamma (*IFNG*) expression in the RNA-seq analysis (*Figure 2—figure supplement 1B*), marked activation of IFN-γ-inducible genes is apparent. We hypothesize that this may be due to perturbations to signaling effectors of the IFN-γ pathway which includes the STAT family of transcription factors that are often aberrantly expressed in leukemia and other cancers (*Caldarelli et al., 2013*; *Spiekermann et al., 2002*; *Muhlethaler-Mottet et al., 1998*). The activation of so many genes involved in antigen processing and presentation as well as macrophage cell surface markers (*Figure 2—figure supplement 1C*) may indicate that these cells are undergoing differentiation toward a more macrophage-like state, consistent with apparent differentiation observed in other DOT1L loss-of-function paradigms (*Bernt et al., 2011*; *Daigle et al., 2011*). By Gene Set Enrichment Analysis (GSEA) (*Subramanian et al., 2005*; *Mootha et al., 2003*), the set of differentially expressed genes were enriched for hematopoietic differentiation factors and anticorrelated with hematopoietic progenitor expression signatures (*Figure 2—figure supplement 1D*). Notably, the cytokine receptors CSF1R and CSF3R, critical signaling inducers of hematopoietic differentiation, were upregulated (*Figure 2—figure supplement 1E*; *Mossadegh-Keller et al., 2013*; *Klimiankou et al., 2017*).

Among the most downregulated genes were many MLL-AF4 target genes (*Kerry et al., 2017*; *Guenther et al., 2008*; *Wilkinson et al., 2013*) including the oncogene FMS-Like Tyrosine Kinase 3

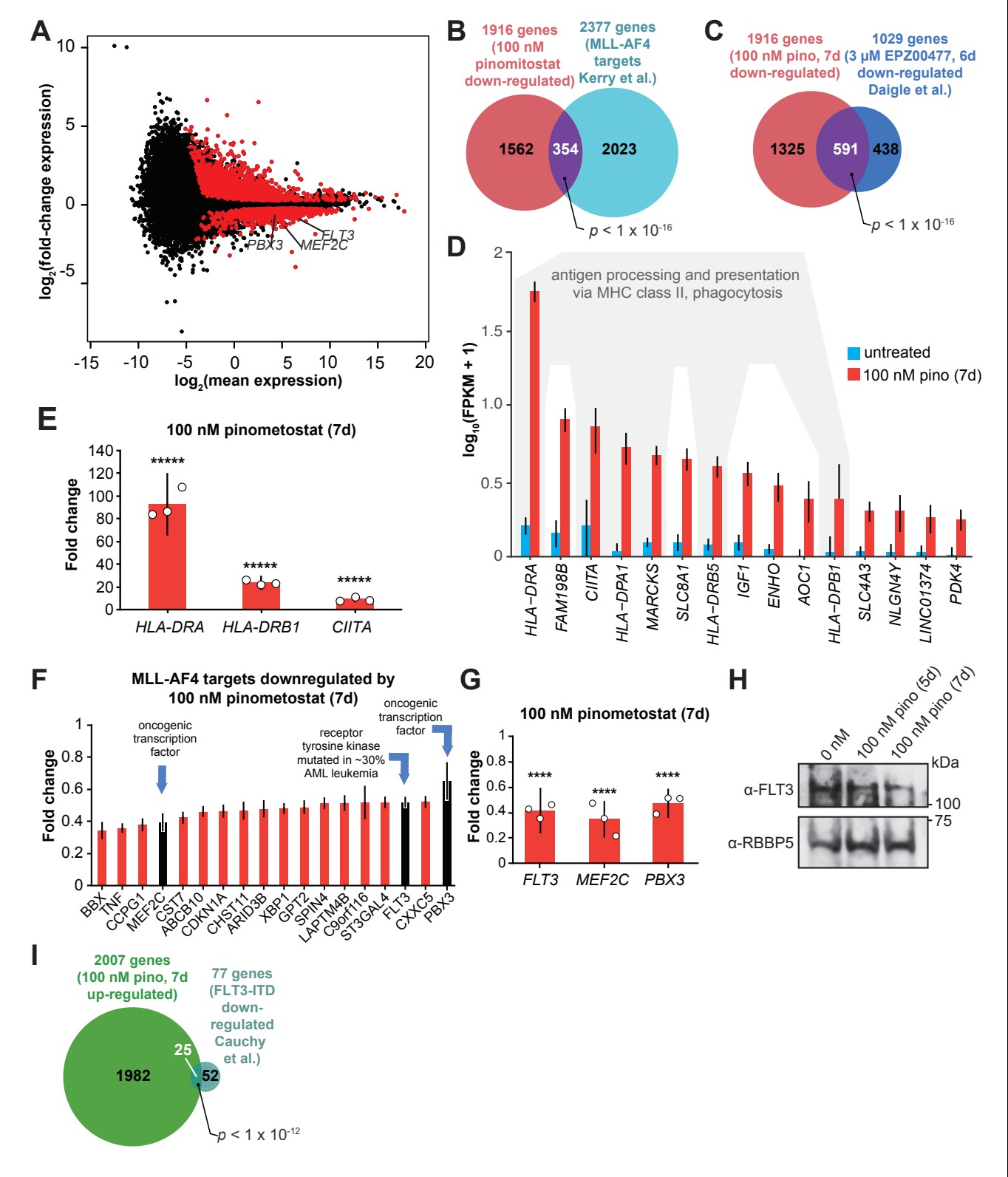

**Figure 2.** DOT1L inhibition downregulates a subset of MLL-AF4 targets including the leukemic oncogene FLT3. (A) MA plot showing genes differentially expressed in MV4;11 cells treated with 100 nM pinometostat or DMSO 7 days as $\log_2$-mean of expression (FPKM) of the DMSO and pinometostat-treated samples versus the $\log_2$-fold change of the mean normalized pinometostat versus DMSO-treated FPKM for three independent replicates. Red represents genes that meet the significance threshold, with an FDR-adjusted $p \leq 0.5$. (B) Venn diagram depicting overlapping genes

*Figure 2 continued on next page*

*Figure 2 continued*

between those downregulated by 100 nM pinometostat and MV4;11 MLL-AF4 targets identified by *Kerry et al., 2017*, p-value computed by two-tailed Fisher Exact test. (C) Venn diagram displaying the overlap between genes downregulated in MV4;11 cells by 100 nM pinometostat treatment (7 days) and treatment with 3 µM of the pinometostat-related compound EPZ004777 for 6 days (*Daigle et al., 2011*). p-Value computed by two-tailed Fisher Exact test. (D) Bar plot depicting upregulated genes with the highest fold changes from RNA-seq analysis of three independent experiments of DMSO- (blue) or pinometostat-treated (red) MV4;11 cells with uncertainty presented as the standard deviation computed by CuffDiff (*Trapnell et al., 2012*) with immune response genes outlined in gray. (E) RT-qPCR analysis showing the fold-change for *HLA-DRA, HLA-DRB1,* and *CIITA* gene expression in MV4;11 cells ± 100 nM pinometostat treatment for 7 days. Results are shown as mean ± S.E.M. of three independent experiments. Student's t-test (*****p ≤ 0.00001). (F) Bar plot depicting the top pinometostat-downregulated genes from the RNA-seq analysis that are previously described MLL-AF4 targets (*Guenther et al., 2008*) including the oncogenes *MEF2C, FLT3,* and *PBX3*. (G) RT-qPCR analysis of *MEF2C, FLT3,* and *PBX3* expression in MV4;11 cells ± with 100 nM pinometostat for 7 days. Results are displayed as mean fold-change ± S.E.M. of three independent experiments; Student's t-test (**** p ≤ 0.0001). (H) Western blot for FLT3 with RBBP5 as loading control in MV4;11 cells treated with 100 nM pinometostat for 5 or 7 days. (I) Venn diagram displaying the overlap between genes upregulated in MV4;11 cells by 100 nM pinometostat treatment (7 days) and genes downregulated in leukemic cells from patients with *FLT3-ITD* vs normal *FLT3* karyotypically normal AML (*Cauchy et al., 2015*). p-Value computed by two-tailed Fisher Exact test.

The online version of this article includes the following figure supplement(s) for figure 2:

**Figure supplement 1.** DOT1L inhibition upregulates components of the interferon gamma pathway and markers of differentiation.

(*FLT3*), the protooncogene Myocyte Enhancer Factor 2C (*MEF2C*), and Pre-B-cell leukemia homeobox 3 (*PBX3*) (*Figure 2F*). These genes all have previously described roles in the development of MLL-rearranged leukemias (*Stubbs et al., 2008*; *Nagel et al., 2017*; *Krivtsov et al., 2006*; *Li et al., 2016*). FLT3 is a receptor tyrosine kinase that regulates proliferation and cell survival via STAT and other signaling pathways. Mutations that constitutively activate FLT3 by internal tandem duplication of its juxtamembrane domain (FLT3-ITD) or point mutations within its kinase domain collectively represent the most frequently occuring genetic lesions in acute myeloid leukemia (*Nagel et al., 2017*; *Levis and Small, 2003*; *Mizuki et al., 2003*). MV4;11 cells are homozygous for the *FLT3-ITD* mutation and highly sensitive to FLT3 inhibition (*Armstrong et al., 2003*; *Levis et al., 2002*). The transcription factor *MEF2C* cooperates with *SOX4* to induce leukemogenesis in mouse models and *MLL-AF9*-expressing hematopoietic progenitors to promote colony formation (*Krivtsov et al., 2006*; *Du et al., 2005*). PBX3 is a transcription factor that acts to stabilize both HOXA9 and MEIS1 localization at a subset of target genes and coexpression of either oncogene with *PBX3* can cause leukemogenesis (*Li et al., 2016*; *Li et al., 2013b*; *Wang et al., 2006*). We verified the reductions in *FLT3, MEF2C* and *PBX3* expression with pinometostat by RT-qPCR and examined FLT3 protein levels by Western blot (*Figure 2G–H*).

We wondered if downregulation of one or more of these genes could be responsible for the reductions in cell proliferation from low-dose pinometostat treatment. Using previously published datasets of MEF2C and FLT3-regulated genes, we first looked at the expression of 15 genes that were downregulated by MEF2C knockout in mouse hematopoietic progenitors (*Stehling-Sun et al., 2009*). Of these genes, only FLT3 was downregulated in our pinometostat-treated cells. Because the expression of nearly all the set of MEF2C-regulated genes was unaffected in our analysis, we moved our focus to FLT3. Previous work by Cauchy et al. identified 138 genes significantly upregulated in karyotypically normal *FLT3-ITD+* AML compared to WT *FLT3* AML patient samples (*Cauchy et al., 2015*). A comparison of those FLT3-ITD-upregulated genes to our pinomeostat downregulated genes yielded a small but significant overlap (*Figure 2—figure supplement 1F*). We saw a more pronounced overlap between genes downregulated in FLT3-ITD+ patient samples and those upregulated by pinometostat, including 10 MHC class II receptors (*Figure 2I*). *PBX3* is the only MLL-AF4 target upregulated in the *FLT3-ITD* samples, suggesting it could be a crucial convergence point of the MLL-AF4 and FLT3-ITD pathways. Collectively, these data suggest that FLT3-ITD may represent an important pathway through which DOT1L inhibition reduces leukemia cell survival. Before delving further into the delineation of the responsible molecular pathways, we first sought to quantitatively define the consequences of low dose DOT1L inhibition on the distribution of the H3K79me2 mark and its causal connection to these gene expression-level changes.

## MLL-AF4 targets downregulated by low dose DOT1L inhibition are highly enriched for H3K79me2

Despite extensive global reductions in H3K79me2 levels, only a subset of MLL-AF4 targets were downregulated by 100 nM pinometostat, necessitating more nuanced measurement of the mark, particularly at MLL-AF4 target genes. The current model, that MLL-AF4 recruits DOT1L to target genes resulting in aberrantly high levels of H3K79me2 and transcriptional activation (*Bernt et al., 2011*; *Guenther et al., 2008*; *Daigle et al., 2013*), has not been rigorously examined by quantitative methods that would be sensitive to small changes. Indeed, the limitations of conventional ChIP-seq preclude unambiguous quantitative analyses for direct comparisons of histone modifications upon global depletion (*Grzybowski et al., 2015*; *Orlando et al., 2014*). To circumvent these problems, we used ICeChIP-seq, a form of native ChIP that uses barcoded internal-standard modified nucleosomes to permit direct quantitative comparison of histone modification density (HMD) at high-resolution across samples (*Grzybowski et al., 2015*; *Grzybowski et al., 2019*; *Shah et al., 2018*).

With ICeChIP we were able to measure a positive correlation ($R^2 = 0.53$) between transcript abundance and H3K79me2 levels in MV4;11 cells (*Figure 3A*), consistent with the speculated role for H3K79me2 in transcriptional activation (*Okada et al., 2005*; *Bernt et al., 2011*; *Chen et al., 2015a*; *Daigle et al., 2011*). However, only 30 of the 250 most highly expressed genes, were downregulated by 100 nM pinometostat treatment, suggesting that H3K79me2 is not necessary to maintain high levels of gene expression at all sites where it is enriched. The genes that were downregulated by 100 nM pinomeostat had higher H3K79me2 levels compared to upregulated genes or all expressed genes, rivalling the most highly expressed genes (*Figure 3B*). Although previous conventional ChIP-seq measurements observed enrichment of H3K79me2 at MLL-fusion target genes (*Bernt et al., 2011*; *Guenther et al., 2008*), our ICe-ChIP-seq analysis revealed equivalent average density at MLL-AF4 targets and the 250 most highly expressed genes (*Figure 3B*). Given that only 12 MLL-AF4 targets are included in that highly expressed gene list, this higher H3K79me2 density is likely due to very efficient recruitment of DOT1L by MLL-AF4 rather than deposition via the transcriptional apparatus (*Schübeler et al., 2004*; *Guenther et al., 2007*). Interestingly, the subset of MLL-AF4 targets that are downregulated by 100 nM pinometostat exhibit still higher levels of H3K79me2 than even MLL-AF4 targets as a whole and appear to be more dependent on H3K79me2 for their expression (*Figure 3A and B*). The only other group of genes analyzed with comparable peak H3K79me2 levels were 'MLL-spreading' genes which display a binding profile that stretches further downstream into the gene body (*Kerry et al., 2017*). The exceptional precision and accuracy of ICeChIP is due to the use of internal calibration standards and is clear from the correlation of HMD measurements of different immunoprecipitation replicates (*Figure 3—figure supplement 1C*). Indeed, H3K79me2 enrichment at these gene groups was remarkably similar in two additional ICeChIP-seq replicates (*Figure 3—figure supplement 1D*).

In all gene categories we examined, 100 nM pinometostat dramatically reduced apparent H3K79me2 density in gene bodies, eliminating the sharp peaks near the TSS and proportionally reducing methylation as it tapers toward the 3' end of the gene body (*Figure 3B*). The upregulated gene set displayed lower-than-average density both before and after treatment, consistent with the transcriptional upregulation occurring as an indirect effect of the dosing. The 100 nM pinometostat downregulated genes, 250 highest expressed genes and MLL-AF4 targets all experienced much higher yet similar reductions in H3K79me2 HMD. The similar reductions in methylation at gene groups that had such different overall responses to gene expression from pinometostat treatment suggests that the expression of some genes is more dependent on H3K79me2-mediated transcriptional activation. Given the modest correlation between H3K79me2 early in the gene body and transcriptional output, we observed an unexpectedly poor linear correlation between fold-change in H3K79me2 HMD versus fold-change in gene expression of differentially expressed genes ($R^2 = 0.13$) (*Figure 3—figure supplement 1B*). However, comparing the absolute differences in HMD to fold-change of gene expression more clearly reveals some interesting trends (*Figure 3C*). Those genes with the largest reductions in HMD (including MLL-AF4 targets) are nearly uniformly downregulated though not in proportion to HMD loss. Conversely, MLL-AF4 targets with smaller HMD reductions are more evenly distributed between both up- and downregulated genes. FLT3-ITD-upregulated genes identified in patient samples (*Cauchy et al., 2015*) have only small reductions in HMD,

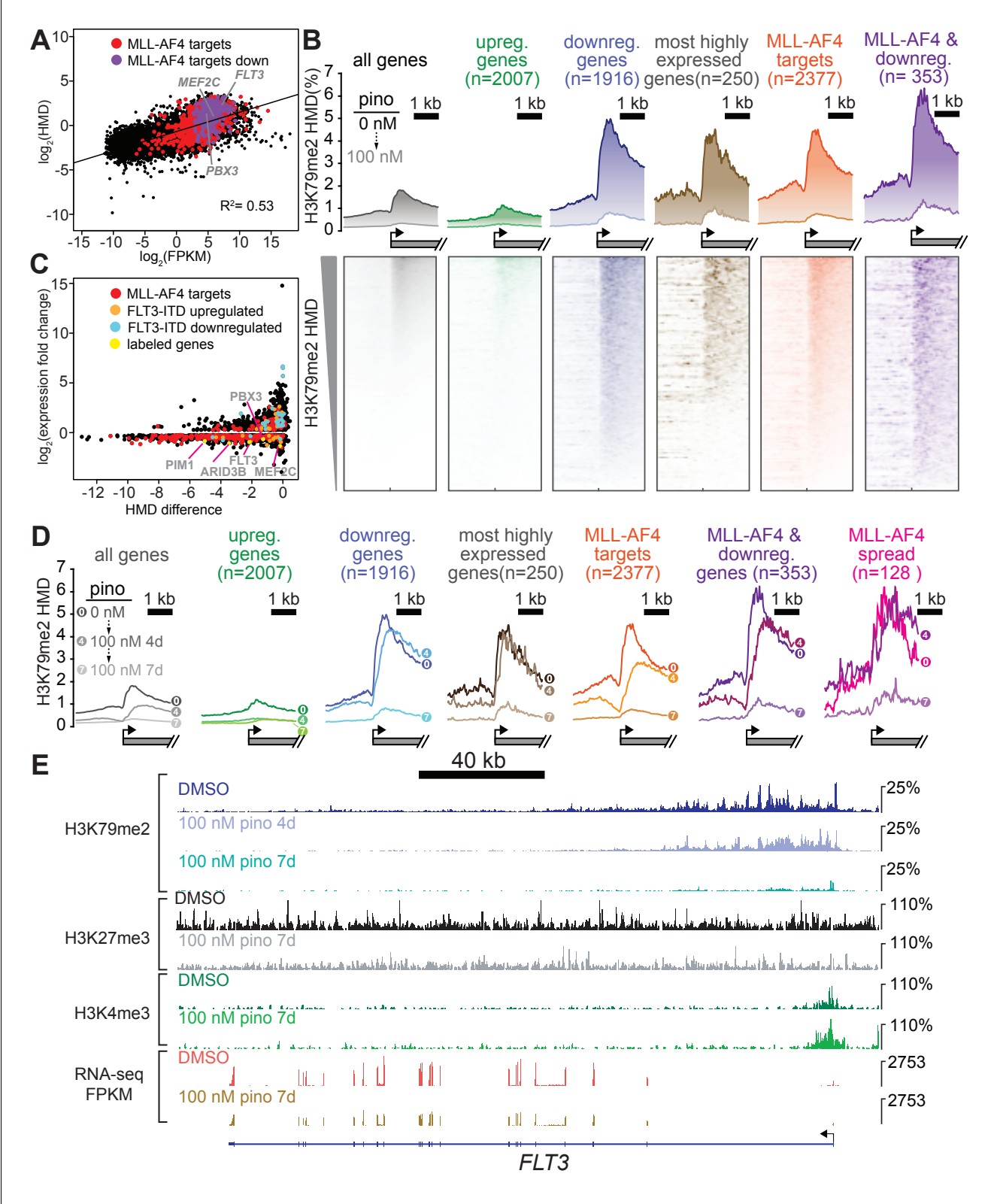

**Figure 3.** Low-dose DOT1L inhibition disrupts H3K79me2 with more pronounced effects on downregulated MLL-AF4 targets. (**A**) Scatterplot of the mean normalized $\log_2$ FPKM (three independent replicates) of genes expressed in DMSO-treated MV4;11 cells plotted versus the $\log_2$ HMD (H3K79me2) for +1000 bp from the TSS. Colors signify: red, MLL-AF4 targets (**Kerry et al., 2017**); purple, MLL-AF4 targets downregulated by 100 nM pinometostat. (**B**) (top) Quantitative measurement of H3K79me2 modification density from ICeChIP-seq of MV4;11 cells treated with 100 nM

*Figure 3 continued on next page*

*Figure 3 continued*

pinometostat for 7 days contoured over the promoters (−2 to +two kp from the TSS) of indicated gene sets, including genes up- or down-regulated by 100 nM pinometostat, the most highly-expressed genes, MLL-AF4 target genes (*Kerry et al., 2017*) as well as those MLL-AF4 targets downregulated by 100 nM pinometostat. (bottom) Heatmaps depicting H3K79me2 density (HMD) for the gene promoter regions shown above ranked by HMD. (C) Scatterplot of genes in MV4;11 cells downregulated by 100 nM pinometostat depicting log$_2$-fold change H3K79me2 HMD (+1000 bp from TSS) versus the log$_2$-fold change of the mean normalized FPKM (three independent replicates) for 100 nM pinometostat or DMSO treated cells. Colors signify: red, MLL-AF4 targets (*Kerry et al., 2017*); orange, FLT3-ITD upregulated genes (*Cauchy et al., 2015*); blue, FLT3-ITD downregulated genes (*Cauchy et al., 2015*); yellow, labeled genes in gray font. (D) H3K79me2 meta promoter profiles as in B, but including curves for 100 nM pinometostat treatment at 4 days, and the promoter set where this complex spreads (*Kerry et al., 2017*). (E) The FLT3 locus as representative of an MLL-AF4 target (*Kerry et al., 2017*; *Guenther et al., 2008*) downregulated by 100 nM pinometostat, displaying MV4;11 ICeChIP-seq tracks for H3K79me2 100 nM pinometostat 4- and 7-day treatment and H3K27me3 and H3K4me3 tracks from 100 nM pinometostat 7-day treatment as well as DMSO control-treated cells and an RNA-seq track (FPKM) from a single replicate of 100 nM pinometostat 7-day treatment and DMSO-treated cells.

The online version of this article includes the following figure supplement(s) for figure 3:

**Figure supplement 1.** H3K79me2 loss is poorly correlated with reductions in gene expression.

suggesting their downregulation is not a direct result of HMD loss but, instead, a secondary effect of FLT3 downregulation.

Interestingly, the MLL-AF4 targets downregulated by low-dose pinometostat (*Figure 2B*) had the largest reductions in H3K79me2 of any gene category examined (*Figure 3B*). These data show that a subset of MLL-AF4 targets have higher levels of H3K79me2 and greater reductions from DOT1L inhibition and are more dependent on this methylation for even moderate levels of expression. Gene expression sensitivity to low-dose DOT1L inhibition may more accurately define 'true' MLL-AF4 target genes whose expression is upregulated by the fusion protein and H3K79me2 hypermethylation than those genes that merely align with MLL1 and AF4 ChIP-seq peaks.

To further define the H3K79me2 depletion trajectory, we also examined the distribution of this modification within gene bodies at an earlier timepoint of pinometostat treatment. Treating MV4;11 cells with 100 nM pinometostat for 4 days had little effect on H3K79me2 HMD at the most highly expressed genes, which likely depend more on DOT1L recruitment by the transcriptional apparatus than by the MLL-fusion protein (*Figure 3D*). Pinometostat treatment for 4 days diminished the 5' H3K79me2 peak at genes downregulated by 7-day pinometostat treatment and at MLL-AF4 targets while only slightly reducing H3K79me2 levels within gene bodies of MLL-AF4 targets. Within the gene bodies of 100 nM (7 day) pinometostat-downregulated genes there was actually an increase in H3K79me2 HMD at the 4-day timepoint. This 3' shift in methylation density away from the transcription start site was even more evident in 'MLL-spreading' genes, which showed little reduction in peak methylation levels seen in other groups. The shifting and near total depletion of H3K79me2 density from 4-day and 7-day 100 nM pinometostat treatment, respectively, is exemplified by several MLL-AF4 target loci (*Figure 6—figure supplement 1D*; *Figure 6—figure supplement 1.F*; *Figure 6—figure supplement 1.G*).

The absence of correlation between H3K79me2 loss and reductions in gene expression suggests that this modification does not have a universal and proportionate effect on gene activation. Rather, it appears some MLL-AF4 targets have higher levels of H3K79me2 and are more sensitive to its depletion. It is possible that the higher methylation levels result in greater dependence on this modification for gene expression at a subset of MLL-AF4 targets. Given the correlation of H3K79me2 depletion with *FLT3-ITD* expression decrements (*Figure 3E*), we next sought to determine if these consequences, were direct, and whether the functional consequences of DOT1L inhibition can be explained by this pathway.

## MLL-r cells with *FLT3-ITD* mutations are hypersensitive to both DOT1L and FLT3 inhibition

As our mechanistic analyses relied on MV4;11 cells (*MLL-AF4*, *FLT3$^{ITD/ITD}$*), we investigated the effects of low dose DOT1L inhibition on three other cell lines to determine whether *FLT3-ITD* could account for increased sensitivity to H3K79me2 ablation. Unlike MV4;11, the MOLM13 cell line harbors an *MLL-AF9* translocation and is heterozygous for the *FLT3-ITD* mutation (*Quentmeier et al., 2003*), lesions that have been shown to cooperate to reduce the latency of leukemia onset in mice (*Stubbs et al., 2008*). We also examined two MLL-translocation cell lines without *FLT3* mutations:

THP-1 (*MLL-AF9*); and SEM (*MLL-AF4*). We note that previous studies of DOT1L inhibitor dosing sensitivity of some MLL-r cell lines (*Daigle et al., 2013*) could be explained by the *FLT3* mutational status, although given the many other genetic background differences in outgrown cell lines it is reasonable that this correlation was not noted.

We treated all four cell lines with 100 nM pinometostat for 7 days. When comparing each cell line to its counterpart with the same MLL-translocation, those with the *FLT3-ITD* mutation were significantly more sensitive to DOT1L inhibition than those with normal *FLT3* alleles (*Figure 4A*, left). After 7 days of 100 nM pinometostat treatment MV4;11 viability was drastically reduced by 74 ± 3% while the viability of SEM, its *MLL-AF4* counterpart with intact *FLT3*, was unaltered within experimental error. MOLM13 viability was somewhat reduced (21 ± 3%) while there was no significant difference in the viability of THP-1 cells. As in MV4;11 cells, MOLM13 cells displayed no change in *HOXA9* or *MEIS1* expression under these conditions (*Figure 4—figure supplement 1A*).

If the heightened sensitivity of MLL-r cell lines to DOT1L inhibition is indeed mediated by reduced *FLT3-ITD* expression then we would expect to see a similar heightened sensitivity to disruption of FLT3 signaling. The small molecule tandutinib (MLN518) inhibits FLT3 kinase activity, severely reducing phosphorylation-mediated activation of downstream targets such as STAT5A (*Clark et al., 2004*). We treated our MLL-r cell lines with 30 nM tandutinib for 7 days. As with the DOT1L inhibition experiments, cell lines with *FLT3-ITD* mutations were significantly more susceptible to the inhibitor's effects (*Figure 4A*, right). Given the variety of other genetic differences amongst these cell lines, these observations can at best be taken as consistent with the hypothesis that the co-occurring *FLT3-ITD* mutations may sensitize MLL-r leukemias to DOT1L inhibition, motivating us to seek more direct examination of FLT3 signaling.

## Impaired FLT3 signaling by DOT1L inhibition culminates in reduced transcription of STAT5A target genes

The *FLT3-ITD* mutation allows FLT3 to phosphorylate STAT5A, a transcription factor that is not activated by wild type FLT3 (*Choudhary et al., 2005*). This aberrant STAT5A phosphorylation licenses translocation to the nucleus to drive target gene transcription, resulting in a hyperproliferative state necessary for leukemia cell survival (*Onishi et al., 1998*; *Choudhary et al., 2007*). We hypothesized that *FLT3-ITD* downregulation by DOT1L inhibition would thereby reduce STAT5A phosphorylation. Indeed, 100 nM pinometostat as well as SGC0946 treatment for 7 days reduced STAT5A phosphorylation in MV4;11 cells without affecting STAT5A protein levels (*Figure 4B* and *Figure 4—figure supplement 1B*), where pinometostat reduced STAT5A phosphorylation by 65 ± 8% (*Figure 4—figure supplement 1C*). We observed that pinometostat treatment slightly reduced STAT5 phosphorylation in MOLM13 cells, consistent with the lower *FLT3-ITD* allele dose, whereas lines with wild type *FLT3* (THP-1, SEM) did not display these effects. As a point of direct comparison, small molecule inhibition of FLT3 signaling yielded markedly reduced STAT5 phosphorylation in lines bearing the *FLT3-ITD* (MV4;11 and MOLM13), with a more modest reduction in SEM cells while phospho-STAT5 was barely detectable in THP-1 cells (*Figure 4B*).

To examine whether FLT3 effects precede other pro-proliferation pathways, we obtained more granular expression kinetics of several downregulated MLL-AF4 targets that have been implicated in leukemogenesis. Expression of *FLT3*, *PBX3*, *PIM1* and *MEF2C* was significantly reduced after 72 hr treatment with pinometostat (*Figure 4C*), however, *FLT3* was the only gene whose expression was reduced 48 hr after treatment, suggesting it is more sensitive to H3K79me2 reductions than the others examined. Though *FLT3* and *MEF2C* are targets of the HOXA9-MEIS1-PBX3 complex, these genes are all targets of the MLL-fusion protein (*Kerry et al., 2017*). The reduction in *FLT3* expression in advance of decreased *PBX3* or *MEF2C* expression lends tentative support to the possibility that DOT1L inhibition directly affects *FLT3* gene expression independently of *PBX3* or *MEF2C*.

Given the early reductions in FLT3-ITD expression and reduced phosphorylation of its target STAT5A, we hypothesized that the pinometostat-induced reductions in proliferation were due to a loss of STAT5A signaling. We performed GSEA (*Subramanian et al., 2005*; *Mootha et al., 2003*) with the pinometostat-downregulated genes and genes upregulated by STAT5A overexpression in human CD34+ hematopoietic progenitors (*Wierenga et al., 2008*) and observed a negative correlation indicative of significant pathway overlap (NES = −1.87, FDR = 0.003, *Figure 4—figure supplement 1D*). We then reexamined our RNA-seq data for previously described STAT5A target genes downregulated by pinometostat and found several, including *PIM1* and *ARID3B* (*Kim et al., 2005*;

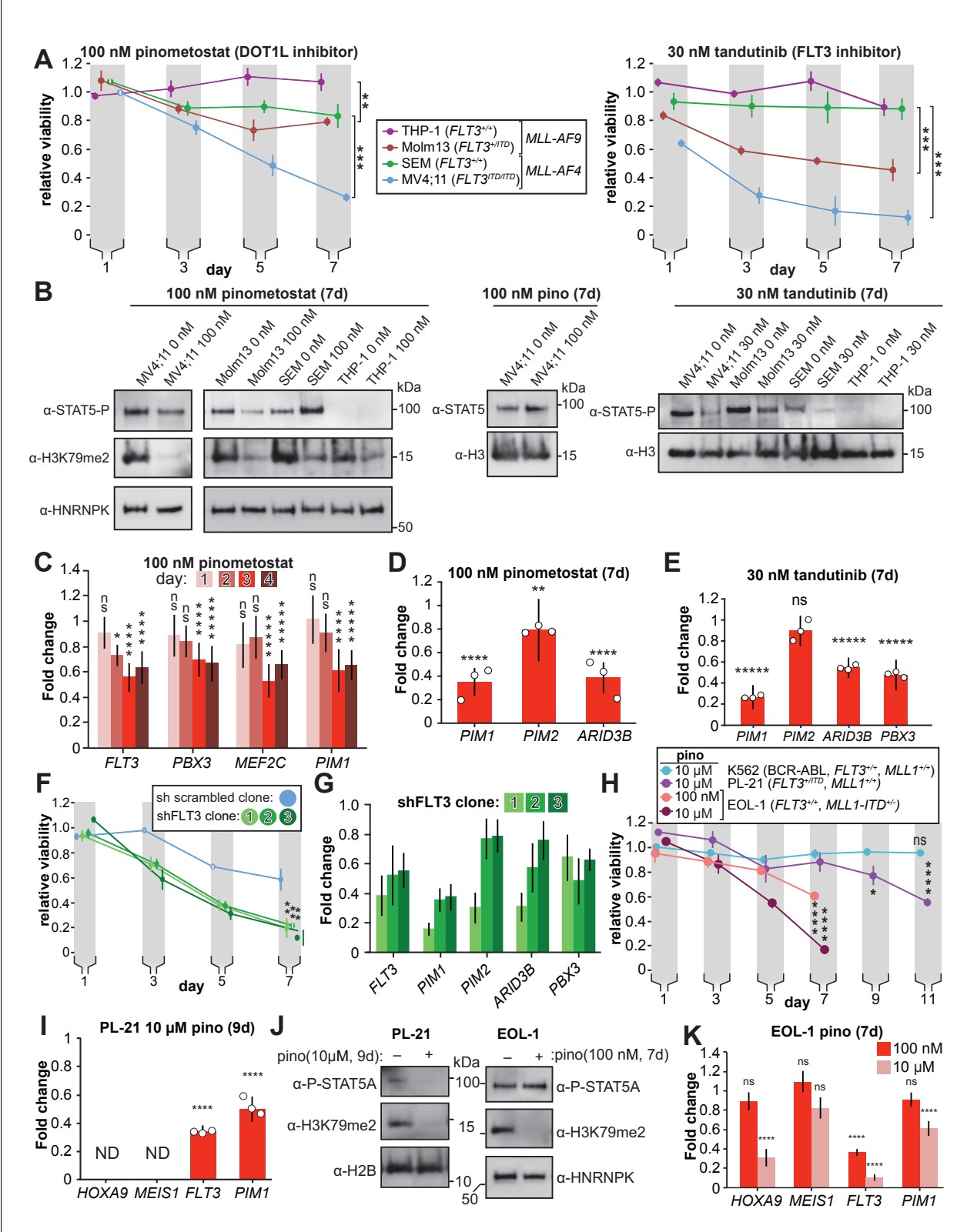

**Figure 4.** DOT1L inhibition reduces STAT5A activation and downregulates STAT5A targets in *FLT3-ITD* leukemia lines. (**A**) MLL-rearranged leukemia lines with genotypes indicated were treated with 100 nM pinometostat (left panel, DOT1L inhibitor) or 30 nM tandutinib (right panel, FLT3 inhibitor MLN518), and relative growth monitored by CellTiter Glo 2.0 assay on the indicated days. Relative viability presented is the mean fraction of luminescence of treated versus side-by-side mock treated cultures (same volume of DMSO) for three independent replicates ± S.E.M. Student's t-test

*Figure 4 continued on next page*

*Figure 4 continued*

(** $p \leq 0.01$, *** $p \leq 0.001$). (B) Western blots of phosphorylated STAT5 (active) or total STAT5A with H3 or HNRNPK as loading controls across the cell lines from panel A treated as indicated; H3K79me2 is monitored in pinometostat-treated lines to confirm inhibition. (C) Time course of gene expression by RT-qPCR, presented as mean fold-change of *FLT3, PBX3, PIM1,* and *MEF2C* in MV4;11 cells ± 100 nM pinometostat at each time point indicated ± S. E.M.; n = 3; Student's t-test (ns $p > 0.05$, * $p \leq 0.05$, **** $p \leq 0.0001$, ***** $p < 0.00001$). (D-E) DOT1L and FLT3 inhibition downregulate STAT5A targets in *FLT3-ITD*. RT-qPCR expression analysis presented as mean fold-change ± S.E.M. for the indicated transcript in MV4;11 cells treated with indicated inhibitor versus mock-treatment for 7 days. Student's t-test (** $p < 0.01$, **** $p < 0.0001$, ***** $p < 0.00001$). (F) Proliferation assay as in panel A, with three clonal populations of MV4;11 cells virally transduced, selected, then induced to express shRNA to FLT3 (*Green et al., 2015*) or a scrambled shRNA (*Yuan et al., 2009*) control by 1 μg/mL doxycycline. Means of fractional viability relative to uninduced cells ± S.E.M. are shown for three independent experiments; Student's t-test (** $p < 0.01$). (G) RT-qPCR analysis of *PIM1, PIM2,* and *ARID3B* expression in MV4;11 cells expressing an inducible shRNA targeting FLT3 (*Green et al., 2015*) for 7 days. Results are depicted as fold-change expression of control cells expressing shRNA to GFP (*Scheeren et al., 2005*). (H) Proliferation assay of K562, PL-21, and EOL-1 cells treated with 10 μM or 100 nM pinometostat using CellTiter Glo 2.0 to measure viability, showing the luminescence fraction of inhibited over DMSO-treated cells. Means ± SE are shown for three independent experiments. Student's t-test of day 7 (EOL-1 cells), day 9 (PL-21), or days 9 and 11 (K562 and PL-21) values: ns $p > 0.05$, * $p < 0.05$, **** $p < 0.0001$. (I) Gene expression analysis by RT-qPCR in PL-21 cells treated for 9 days with 10 μM pinometostat. Results are displayed as fold-change over DMSO-treated cells with means ± SE for three independent experiments (ND = not detected). Student's t-test (**** $p < 0.0001$). (J) Western blots of (left) cell extract from PL-21 cells treated with 10 μM pinometostat 9 days and (right) EOL-1 cells treated with 100 nM pinometostat for 7 days and then blotted for H3K79me2 and p-STAT5 with H2B or HNRNPK as a loading controls. (K) Gene expression analysis by RT-qPCR in EOL-1 cells treated for 7 days with 100 nM or 10 μM pinometostat. Results are displayed as fold-change over DMSO-treated cells with means ± SE for three independent experiments. Student's t-test (ns $p > 0.05$, **** $p < 0.0001$).

The online version of this article includes the following figure supplement(s) for figure 4:

**Figure supplement 1.** FLT3 knockdown reduces STA5A activation.

*Ribeiro et al., 2018*; *Figure 4—figure supplement 1E*). The PIM proteins are a family of 3 protoon-cogene serine/tyrosine kinases (PIM1-3) that are upregulated in, and indicative of poor prognosis in leukemia, prostate, mesothelioma and other cancers (*Mizuki et al., 2003*; *Kim et al., 2005*; *Amson et al., 1989*; *Cibull et al., 2006*; *Peltola et al., 2009*; *Deneen et al., 2003*). However, only *PIM1* and *PIM2* expression is increased in FLT3 inhibitor-resistant *FLT3-ITD* patient samples and exogenous expression of either *PIM1* or *PIM2* can rescue proliferation defects caused by loss of FLT3 activity in MOLM14 cells (*MLL-AF9*, *FLT3-ITD* heterozygous) (*Adam et al., 2006*; *Green et al., 2015*; *Fathi et al., 2012*). Although *PIM1* and *PIM2* are both downregulated in our RNA-seq analysis (*Figure 4—figure supplement 1E*), we observed a much greater reduction in *PIM1* expression by RT-qPCR (*Figure 4D*). Similar gene expression changes were also observed with a different DOT1L inhibitor (*Figure 4—figure supplement 1B*). Treating MV4;11 cells with tandutinib (FLT3 inhibitor) resulted in downregulation of *PIM1, ARID3B,* and *PBX3* but not *PIM2* (*Figure 4E*). Treating MOLM13 cells with pinometostat also reduced expression of *MEF2C, FLT3* and *PIM1*, but caused no change in *PBX3* expression (*Figure 4—figure supplement 1F*).

If the FLT3 and DOT1L inhibitors have overlapping functions through inhibition or downregulation of *FLT3*, respectively, then we could potentially observe synergy in the effects on MV4;11 proliferation if we treated with both inhibitors simultaneously. We performed a coarse analysis using a combination of inhibitors at concentrations that individually have modest effects on proliferation to examine if they might produce a greater effect on viability when combined (*Figure 4—figure supplement 1G*). We also treated MV4;11 cells with the PIM1 inhibitor quercetegenin and observed increased toxicity when combining the inhibitors on day 5 that was overtaken by pinometostat only at day 7 when cell viability is very low for both treatments (*Figure 4—figure supplement 1G*). The DOT1L inhibitor has a delayed effect compared to the PIM1 and FLT3 inhibitors, which complicates comparisons, but nonetheless, through our coarse analysis of one set of concentrations for both inhibitors we observed small but significant differences in proliferation when using inhibitors singly or in combination.

To directly interrogate the effects of *FLT3* on MLL-r leukemia proliferation without complications from different genetic backgrounds, we used viral transduction to insert a tet-inducible shRNA targeting *FLT3* into MV4;11 cells. With modest knockdown of *FLT3* (*Figure 4G*), we observed significant reductions in the proliferation of three different clonal lines as compared to a scrambled shRNA (*Figure 4F*). *FLT3* knockdown reduces MV4;11 proliferation and STAT5A phosphorylation (*Figure 4—figure supplement 1I*), analogous to the effects of pinometostat treatment. Akin to the DOT1L and FLT3 inhibitors (*Figure 4D–E*), *FLT3* knockdown also significantly reduced the expression of the

STAT5A target genes *PIM1* and *ARID3B*, with *PIM2* expression reduced in only 1 of 3 clones (*Figure 4G*). We observe more modest reduction in STAT5A signaling upon FLT3-ITD knockdown compared to pinometostat treatment, perhaps indicating that other kinases such as JAK1-3 or TYK2, with previously observed roles in STAT5A activation, may also activate STAT5A in this context and further, that the function of one or more of these other kinases may also be reduced by pinometostat treatment (*Paukku and Silvennoinen, 2004*). Interestingly, *FLT3* knockdown also resulted in *PBX3* downregulation, suggesting that FLT3 can regulate the expression of this oncogenic transcription factor, in line with previous observations (*Cauchy et al., 2015*). Collectively, these data suggest that the DOT1L inhibitors may act, in part, by disrupting FLT3 signaling culminating in a reduction in STAT5A target expression and function.

## DOT1L inhibition reduces proliferation in MLL-PTD and non-MLL-rearranged FLT3 mutant leukemia

The pronounced sensitivity of the FLT3-ITD/STAT5A signaling axis to DOT1L inhibition raises the possibility that non-MLL-rearranged leukemias with FLT3-ITD mutations, representing 30–40% of acute myeloid leukemias, may also be susceptible to DOT1L inhibition. We observed a reduction in the viability of the *FLT3-ITD* heterozygous, non-MLL-rearranged leukemia cell line PL-21 after treatment with 10 μM pinometostat for 9 days (*Figure 4H*), accompanied by a reduction in both *FLT3* and *PIM1* expression (*Figure 4I*). Whereas the viability of K562 cells, an erythroleukemic cell line with a BCR-ABL translocation was not affected after treatment with 10 μM pinometostat for 11 days. We observed reductions in H3K79me2 and STAT5A phosphorylation after 9 days pinometostat treatment, suggesting that DOT1L inhibition may also reduce the viability of non-MLL-rearranged FLT-ITD leukemias through disruption of FLT3-ITD/STAT5A signaling (*Figure 4J*). Furthermore, expression of the MLL-r leukemic drivers *HOXA9* and *MEIS1* were not detectable by RT-qPCR in PL-21 cells (*Figure 4I*), arguing that FLT3-ITD targeting by pinometostat is completely distinct.

In a previous study, DOT1L inhibition in leukemia cell lines with MLL1 partial tandem duplications (MLL-PTD) reduced cell viability, downregulated MLL1 target genes including *HOXA9* and induced apoptosis and differentiation (*Kühn et al., 2015*). We recapitulate these findings with 10 μM pinometostat reduced EOL-1 (MLL-PTD, intact *FLT3*) proliferation and *HOXA9* expression, noting also reductions in *FLT3* and *PIM1* expression but no change in *MEIS1* expression (*Figure 4H,K*). Surprisingly, a 10-fold lower dose of pinometostat also reduced EOL-1 viability (*Figure 4H*) and *FLT3* expression with no discernable changes in *HOXA9*, *MEIS1* or *PIM1* expression (*Figure 4K*). This finding is congruent with and potentially explains how low-dose pinometostat treatment was able to reduce the viability of EOL-1 xenografts in rats without affecting *HOXA9* expression (*Kühn et al., 2015*). There was no observable reduction in STAT5A phosphorylation after 7 days 100 nM pinometostat (*Figure 4J*), consistent with previous studies showing that WT *FLT3* has little effect on STAT5A activation (*Choudhary et al., 2005*). WT *FLT3* is typically upregulated by MLL-fusions and is able to activate other pathways involved in cell growth and proliferation such as PI3K/AKT (*Armstrong et al., 2002*; *Cauchy et al., 2015*; *Choudhary et al., 2005*) One or more of these FLT3-activated growth signaling pathways may be essential for EOL-1 and MLL-PTD leukemia cell survival just as the STAT5A signaling pathway appears to be for FLT3-ITD leukemia. To further interrogate the H3K79me2-dependence of leukemia survival on FLT3-ITD/STAT5A signaling, we sought to ectopically restore this signaling pathway upon DOT1L inhibition to potentially rescue viability.

## Overexpression of constitutively active STAT5A rescues proliferation and reductions in gene expression caused by DOT1L inhibition

Unfortunately, overexpression of *FLT3-ITD* for an attempted rescue of DOT1L inhibition proved technically challenging, as retrovirally introduced ectopic expression was rapidly silenced or dropped out during selection as has been observed in other contexts (*Spiekermann et al., 2002*). To further interrogate this pathway's functional significance, we sought to perturb signaling downstream of FLT3-ITD via STAT5A alterations.

To potentiate STAT5A activity, we overexpressed a constitutively active STAT5A mutant to examine whether this could counteract the reduction of upstream FLT3-ITD levels by DOT1L inhibition. STAT5A is 'activated' through phosphorylation at multiple sites, facilitating translocation into the nucleus and activation of gene targets. Previous work showed that H299R and S711F mutations

create a constitutively active murine *Stat5a* able to activate target genes independently of upstream signaling (*Onishi et al., 1998*), which phenocopies the effects of exogenous *FLT3-ITD* expression including hyperproliferation and inhibition of myeloid maturation (*Moore et al., 2007*). We used a lentiviral system to generate individual MV4;11 clonal cell lines with stably incorporated, inducible human *STAT5A* mutated at the corresponding residues H298R and S710F (*STAT5A-CA*), all of which exhibit several-fold induction with doxycycline (*Figure 5A* and *Figure 5—figure supplement 1A*). Ectopic expression of *STAT5A-CA* was able to partially rescue proliferation when challenged with 30 nM FLT3 inhibitor tandutinib, confirming the capacity of this mutant to complement impaired FLT3-ITD signaling (*Figure 5—figure supplement 1B*).

Remarkably, *STAT5A-CA* overexpression also rescued pinometostat-induced proliferation reductions (*Figure 5B*) in proportion to each clone's *STAT5A-CA* expression level (*Figure 5A*). Similar results were also observed with a different DOT1L inhibitor (*Figure 5—figure supplement 1E*). Clone three was unable to rescue proliferation substantially, perhaps because it had the lowest expression of *STAT5A/STAT5A-CA* (*Figure 5A* and S5E). As another control, we similarly overexpressed MEF2C, yet it displayed no effect on the viability of MV4;11 cells treated with 100 nM pinometostat (*Figure 5—figure supplement 1C*).

To gain a molecular understanding of how ectopic *STAT5A-CA* expression could rescue proliferation of inhibitor-treated cells, we measured expression of the STAT5A targets *PIM1*, *PIM2*, and *ARID3B* by RT-qPCR. Expression of *STAT5A-CA* restored expression of *PIM1*, *PIM2*, and *ARID3B* in both DOT1L inhibitor- and FLT3 inhibitor-treated MV4;11 cells (*Figure 5C* and *Figure 5—figure supplement 1D*).

Because ectopic expression of *STAT5A-CA* is able to rescue proliferation of MV4;11 cells and the expression of STAT5A targets including the anti-apoptotic *PIM1* oncogene, we examined whether *STAT5A-CA* overexpression could rescue MV4;11 cells from apoptosis. A previous study observed that ~30% of MV4;11 cells treated with 1 µM pinometostat for 6 days were undergoing apoptosis (*Daigle et al., 2013*). We analyzed apoptosis in MV4;11 cells treated with increasing concentrations of pinometostat for 7 days (*Figure 5D and E*). We observed 25.5 ± 0.3% apoptotic cells when treating with 1 µM pinometostat and a still sizeable proportion (15 ± 1%) of apoptotic cells when treating with just 100 nM pinometostat. Yet upon treatment of STAT5A-CA clone 1 with 100 nM pinometostat for 7 days, we observed no significant induction of apoptosis as compared to the DMSO control (*Figure 5D and E*). Thus, we concluded that recovering STAT5A function can rescue MV4;11 cells from apoptosis induced by 100 nM pinometostat. It is striking that despite marked gene expression changes caused by low-dose DOT1L inhibition, one signaling pathway, FLT3-ITD to STAT5A, is able to account for the bulk of the phenotypic and molecular changes we measured. Given that the rescue was nevertheless incomplete, we investigated other potential secondary contributors to the proliferation and gene expression consequences of low-dose DOT1L inhibition.

## An ancillary DOT1L-dependent pathway limits proliferation through PRC2 signaling

Although H3K79me2 potentiates transcription, our RNA-seq analysis revealed the upregulation of thousands of genes when treating with pinometostat. One potential explanation for this effect is the downregulation of the repressive PRC2 complex members *EZH2* and *EED* and consequent reductions in global levels of the transcriptionally repressive H3K27me3 mark (*Figure 6A–B*, *Figure 6—figure supplement 1A*). PRC2 deposits the facultative heterochromatin H3K27me3 modification and, although antagonistic to MLL1 and H3K4me3 deposition (*Kim et al., 2013*), is necessary for MLL-r leukemogenesis (*Shi et al., 2013*; *Zhou et al., 2011*; *Neff et al., 2012*). Analysis by quantitative ICeChIP revealed that 100 nM pinometostat decreased H3K27me3 genome-wide (*Figure 6C*). Promoter H3K27me3 levels are reduced by 2–5% on average with more pronounced decreases observed among downregulated genes and MLL-AF4 targets than upregulated or all genes (*Figure 6C*). However, H3K27me3 levels in untreated cells were much higher in pinometostat-upregulated genes, perhaps indicating that these genes are more reliant on PRC2 to buffer their expression. H3K27me3 levels are lower throughout gene bodies in DOT1L inhibited cells, as apparent at individual loci (*Figure 3E*, *Figure 6—figure supplement 1D-G*). Analysis by ICeChIP-qPCR of H3K27me3 through two additional independent experiments focusing on representative promoters of genes both up- and downregulated by 100 nM pinometostat revealed methylation reductions consistent with the sequencing data (*Figure 6—figure supplement 1H*).

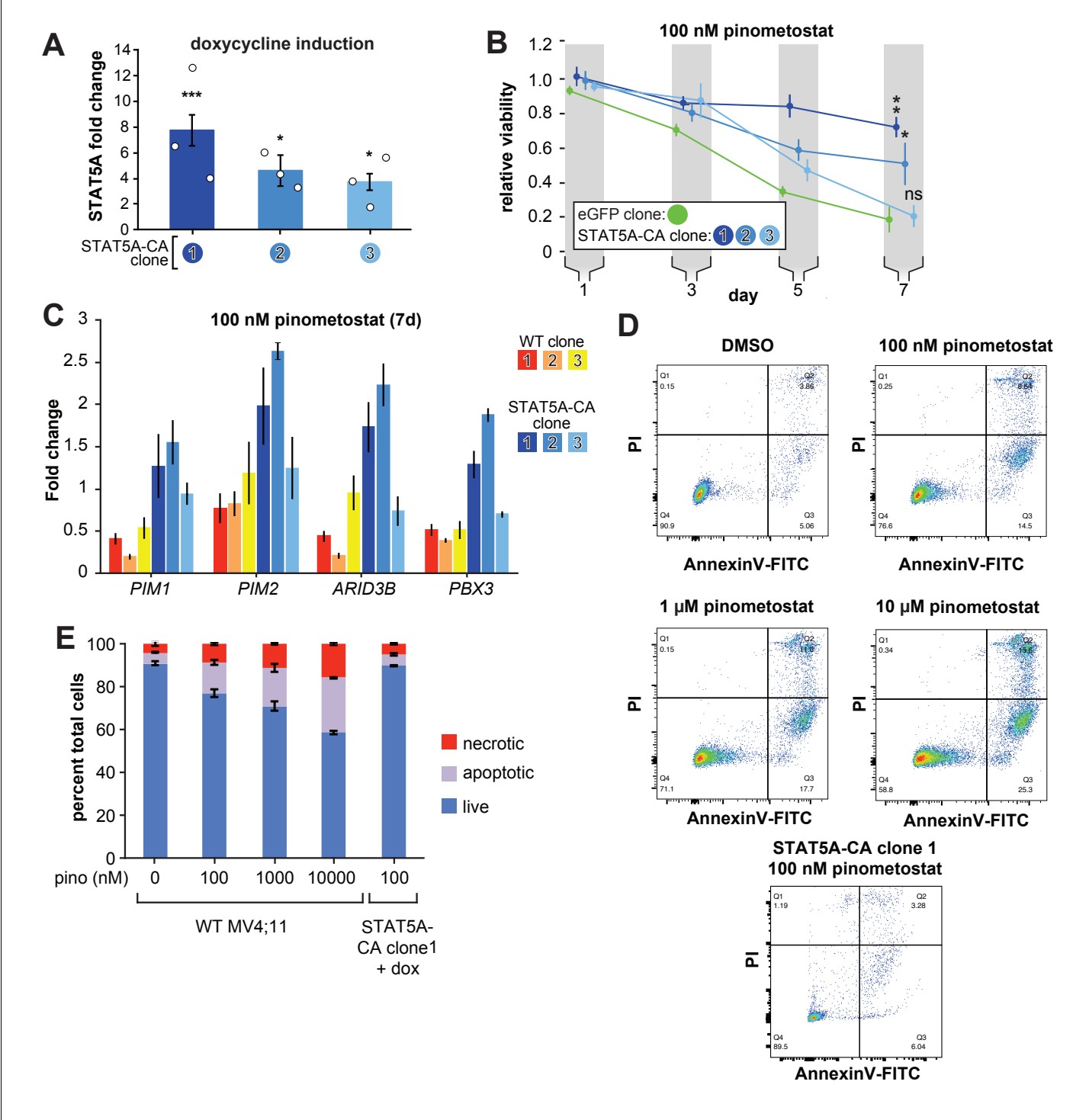

**Figure 5.** Exogenous expression of constitutively active *STAT5A* partially rescues proliferation and gene expression effects of DOT1L inhibition. (**A**) RT-qPCR analysis of *STAT5A* expression from three monoclonal isolates of MV4;11 cells virally transduced with a tet-inducible constitutively active *STAT5A* (*STAT5A-CA*) depicted as fold-change over untransduced cells with standard error of the mean. Student's t-test (* $p < 0.05$, *** $p < 0.001$). (**B**) Proliferation assay of MV4;11 clonal isolates from panel A. induced to express *STAT5A-CA* or eGFP with 1 µg/mL doxycycline and treated concomitantly with 100 nM pinometostat. We determined the fractional viability of each clone as the luminescence from a CellTiter Glo 2.0 assay with pinometostat-treatment normalized to DMSO-treated cells, both induced to express *STAT5A-CA or eGFP*, to accommodate for any potential increases in viability. Means ± SE are shown for three independent experiments with Student's t-test for day 7 values (**** $p \leq 0.0001$). (**C**) Gene expression analysis by RT-qPCR of STAT5A target genes in WT MV4;11 cells or MV4;11 *STAT5A-CA* clones from A. induced with 1 µg/mL doxycycline and treated

*Figure 5 continued on next page*

*Figure 5 continued*

with 100 nM pinometostat for 7 days. Results are displayed as fold-change over DMSO-treated WT cells. (**D**) Quantitative measurement by flow cytometry of live, apoptotic (Annexin V-FITC) and necrotic cells (propidium iodide) of WT MV4;11 cells or cells exogenously expressing *STAT5A-CA* (clone 1) and treated with increasing concentrations of pinometostat. Images of gated FITC vs. PI signal are shown for one of three independent experiments, with all replicates quantified in the bar plot in **E**.

The online version of this article includes the following figure supplement(s) for figure 5:

**Figure supplement 1.** *STAT5A-CA* overexpression partially rescues proliferation and gene expression effects caused by FLT3 inhibition.

We next sought to interrogate the functional impact of the PRC2 signaling axis by experimental perturbation. As PRC2 is necessary for repression of *IFNG* (IFN-γ) and proper differentiation in T-cells (*Borkin et al., 2015*), we wondered if the upregulated genes found in our RNA-seq analysis, many of which are components of the IFN-γ-response, were upregulated as a result of a loss of H3K27me3-mediated repression. To investigate this possibility, we treated MV4;11 cells with 10 μM EI1 EZH2 inhibitor (*Ueda et al., 2014*) and observed dramatically reduced global H3K27me3 (*Figure 6D*) and proliferation (*Figure 6—figure supplement 1B*), consistent with previously observed sensitivities of MOLM13 and MV4;11 (*Ueda et al., 2014*). EI1 treatment had comparatively little effect on the class of genes massively overexpressed during DOT1L inhibition (*Figure 6E*, compare to *Figure 2D and E*). Surprisingly, EZH2 inhibition downregulated *HOXA9* and *MEIS1* expression (which only occurs with higher doses of pinometostat *Daigle et al., 2013*), with no changes in *FLT3* expression (*Figure 6F*) or STAT5 phosphorylation (*Figure 6—figure supplement 1C*). The greater reduction in global H3K27me3 from 10 μM EI1 than 100 nM pinometostat may account for the lack of effect on *HOXA9* and *MEIS1* expression by pinometostat. Collectively, these data argue that the PRC2 pathway is largely independent of the FLT3-ITD-STAT5A pathway, culminating in distinct target gene expression consequences, that may converge for only a few targets, such as *PIM1* and *ARID3B*.

Next, we queried the functional consequences of rescuing EZH2 expression in the context of low-dose DOT1L inhibition. Inducible overexpression of *EZH2* was only able to partially rescue proliferation in M4;11 cells treated with pinometostat, suggesting that a small portion of the effects on MV4;11 viability is due to reduced PRC2 function (*Figure 6G*). The nearly complete rescue from intervening in the FLT3-ITD-STAT5A pathway compared to the modest rescue from PRC2, suggests that the former is the predominant source of pinometostat-induced effects on proliferation in this leukemia background.

## STAT5A-CA overexpression rescues the viability of MV4;11 cells treated with MLL1 inhibitors

Our observations suggest that most of the toxicity from low-dose DOT1L inhibition in MLL-r, *FLT3-ITD+* leukemia cell lines stems from downregulation of *FLT3* and subsequent loss of STAT5A phosphorylation. We wanted to know if this effect was specific to H3K79me2 depletion, or attributable to disruption of MLL-fusion-induced gene activation. To distinguish between these two mechanisms, we employed small-molecule MLL1 inhibitors, potent and effective treatments for MLL-r leukemia (*Cao et al., 2014*; *Borkin et al., 2015*), as orthologous means of disrupting MLL-fusion function. These compounds inhibit MLL1 in different ways but both disrupt the leukemic gene expression profile, specifically downregulating the oncogenes *HOXA9*, *MEIS1*, *FLT3* and *BCL2* (*Cao et al., 2014*; *Borkin et al., 2015*). MI-503 competitively antagonizes binding of MENIN to MLL1, an interaction that is necessary for MLL-fusion complex localization to target genes and leukemogenesis (*Borkin et al., 2015*; *Li et al., 2013a*; *Yokoyama et al., 2005*). Another small molecule, MM-401 inhibits the methyltransferase activity of MLL1 by disrupting its interaction with WDR5, a complex member necessary for full enzymatic activity of MLL1 but not MLL2-4 or SET1 complexes (*Cao et al., 2014*). We treated MLL-r cell lines with low concentrations of MI-503 or MM-401 and observed greater reductions in the proliferation of *MLL-r*, *FLT3-ITD+* cells than their WT *FLT3* counterparts (*Figure 7A and B*).

Given the similar effects of DOT1L and MLL1 inhibitors on MLL-r cell proliferation and gene expression, that both histone modifications are involved in transcriptional activation and the extensive literature describing dynamic cross-talk between chromatin modifications (*Kim et al., 2013*;

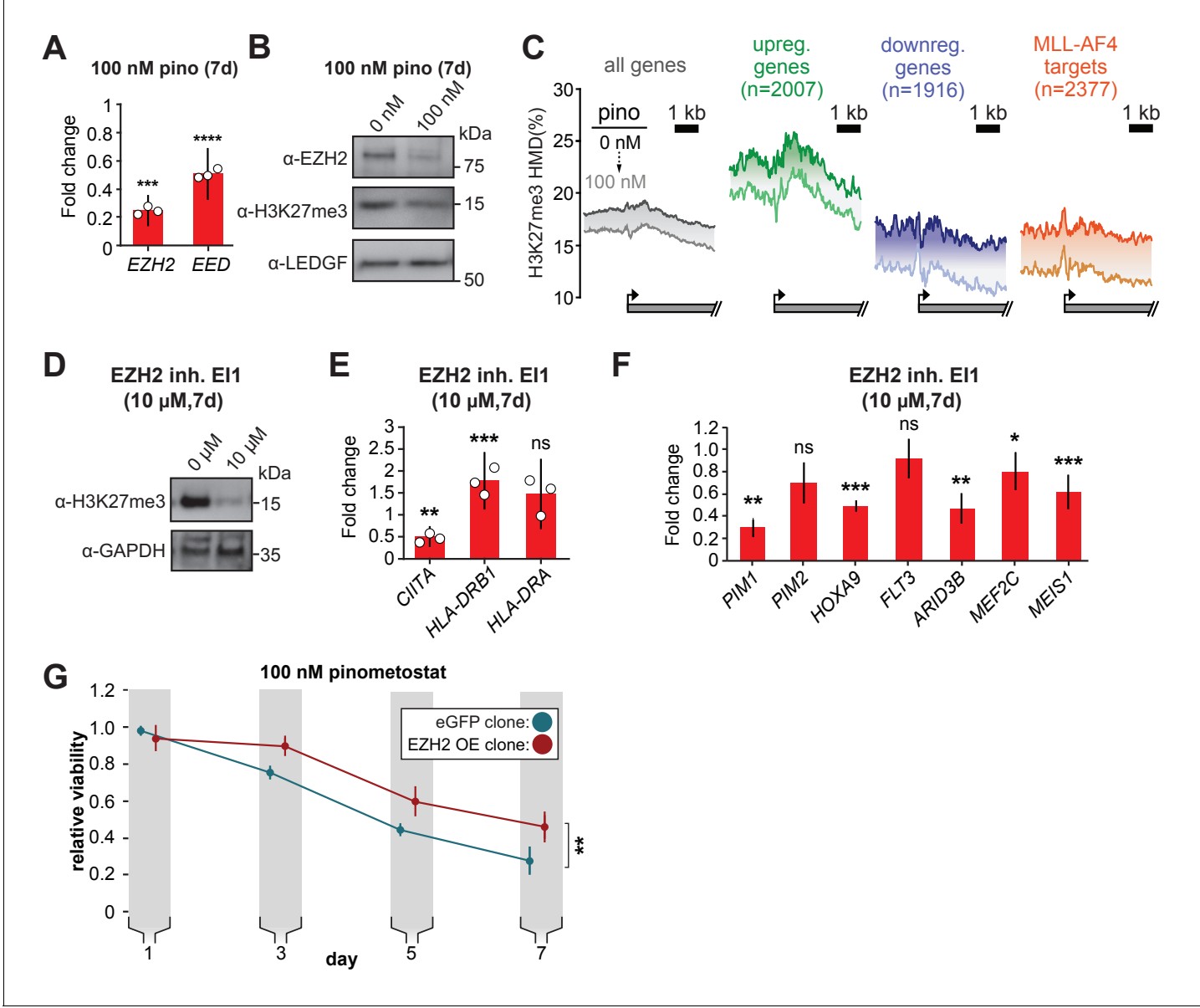

**Figure 6.** PRC2 function is an ancillary pathway dependent on DOT1L and necessary for leukemia proliferation. (A) RT-qPCR analysis of the components of the polycomb complex *EZH2* and *EED* expression in MV4;11 cells ± 100 nM pinometostat for 7 days. Results are displayed as mean fold-change vs. DMSO-treated cells ± S.E.M. of three independent experiments. Student's t-test (*** p < 0.001, **** p < 0.0001). (B) Western blot of EZH2, H3K27me3 and LEDGF as loading control in MV4;11 cells treated ± 100 nM pinometostat for 7 days. (C) Quantitative ICe-ChIP-seq from MV4;11 cells treated with 100 nM pinometostat for 7 days displaying H3K27me3 histone methylation density contoured over promoters from −2000 to +4000 of the TSS of either all expressed genes, genes up- or downregulated by 100 nM pinometostat or MLL-AF4 target genes (*Kerry et al., 2017*). (D) Western blot for H3K27me3 with GAPDH as a loading control in MV4;11 cells treated with EI1 for 7 days. (E) RT-qPCR analysis of MHC class II genes and master regulator *CIITA* expression from MV4;11 cells ± 10 µM EZH2 inhibitor EI1. Results are displayed as mean fold-change vs. DMSO-treated cells ± S.E.M. of three independent experiments. Student's t-test (** p = 0.01, *** p = 0.001). (F) Fold change of RT-qPCR analysis of gene expression MV4;11 cells ± 10 µM EZH2 inhibitor EI1. Results are the average three independent experiments ± S.E.M. Student's t-test (* p < 0.05, ** p < 0.01, *** p = 0.001). (G) Proliferation assay of MV4;11 cells virally transduced with tet-inducible EZH2 or eGFP treated with 100 nM pinometostat and induced with 1 µg/mL doxycycline to express EZH2 or eGFP for 7 days showing the luminescence fraction of inhibited over uninhibited from a CellTiter Glo 2.0 assay. Means ± SE are shown for three independent experiments. Student's t-test of day 7 values (** p = 0.01).

The online version of this article includes the following figure supplement(s) for figure 6:

**Figure supplement 1.** PRC2 inhibition reduces leukemia survival without affecting STAT5A actiation.

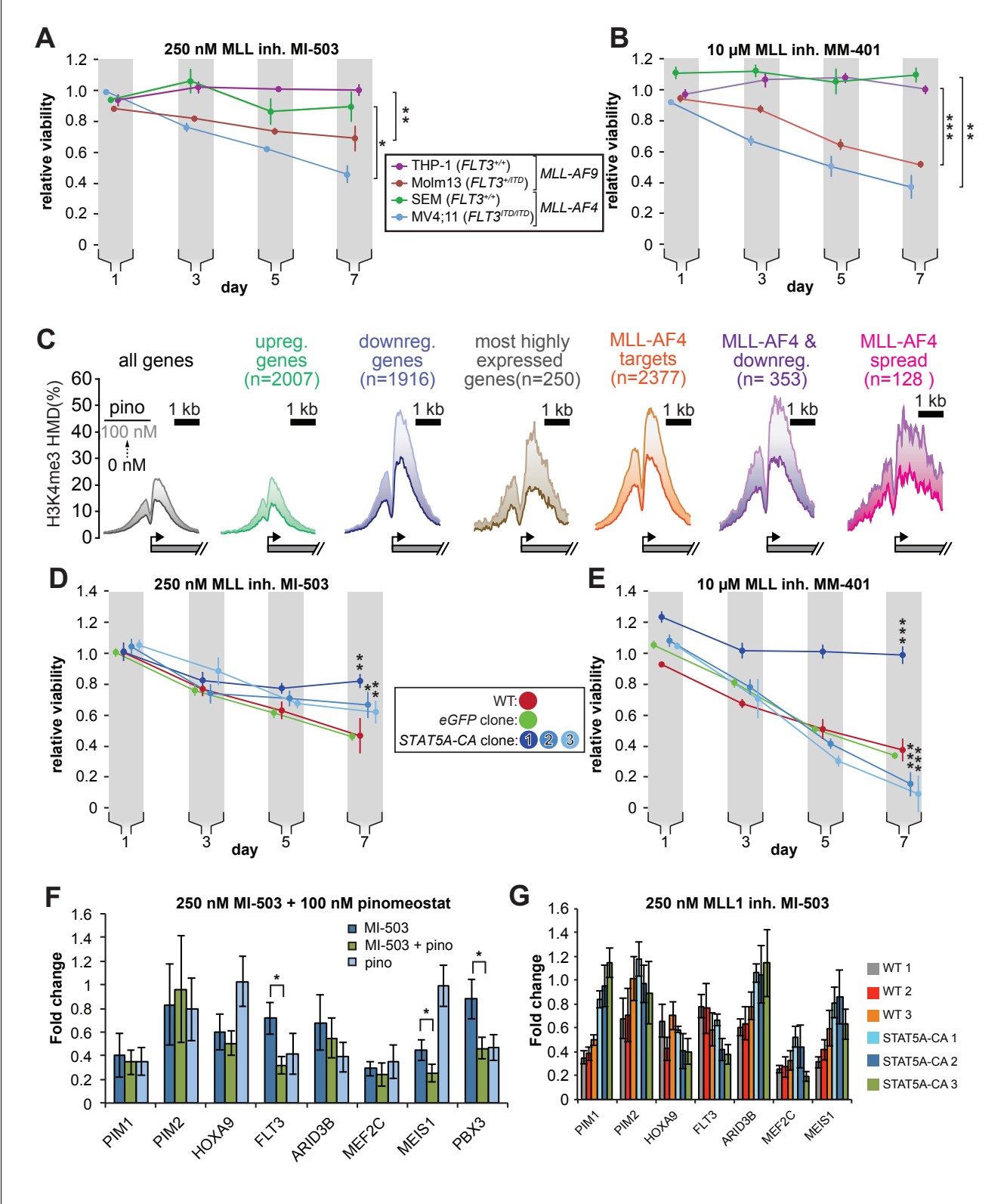

**Figure 7.** STAT5A-CA overexpression rescues the viability of MV4;11 cells treated with MLL1 inhibitors. Proliferation assay of MLL-r cell lines treated with (A) 250 nM MI-503 (MLL1-Menin interaction inhibitor) or (B) 10 μM MM-401 (MLL1 histone methyltransferase inhibitor) for 7 days. Viability was measured by CellTiter Glo 2.0 assay and results are displayed as the fraction of luminescence of inhibitor-treated over DMSO-treated cells. Means ± SE are shown for three independent experiments. Student's t-test (* p < 0.05, ** p < 0.01, *** p < 0.001). (C) H3K4me3 histone methylation density from
*Figure 7 continued on next page*

*Figure 7 continued*

−2000 to +2000 of the TSS from quantitative ICeChIP-seq from MV4;11 cells treated with 100 nM pinometostat for 7 days for genes up- or downregulated by 100 nM pinometostat, the most highly expressed genes, MLL-AF4 target genes (*Kerry et al., 2017*) as well as those MLL-AF4 targets downregulated by 100 nM pinometostat. (D and E) Proliferation assay of MV4;11 *STAT5A-CA* clonal isolates induced to express *STAT5A-CA* or eGFP with 1 μg/mL doxycycline and treated with D. 250 nM MI-503 or (E) 10 μM MM-401. Viability was measured and results displayed as in A and B. Means ± SE are shown for three independent experiments. Student's t-test (* p < 0.05, ** p < 0.01, *** p < 0.001). (F) Gene expression analysis by RT-qPCR of MLL-fusion and STAT5a targets in MV4;11 cells treated with 250 nM MI-503 MLL1 inhibitor, 100 nM pinometostat DOT1L inhibitor or a combination for 7 days. Means ± S.E.M. are shown for three independent experiments (* p < 0.05). (G) Gene expression analysis by RT-qPCR of MLL-fusion and STAT5A targets in WT and STAT5A-CA MV4;11 cells treated with 250 nM MI-503 MLL1 inhibitor for 7 days. Means ± S.E.M. are shown for technical replicates of individual experiments.

The online version of this article includes the following figure supplement(s) for figure 7:

**Figure supplement 1.** MLL1 inhibitors reduce STAT5A phosphorylation.

---

*Chen et al., 2015b*; *Schmitges et al., 2011*; *Voigt et al., 2012*) we were curious as to how perturbations in H3K79 methylation might affect the distribution of H3K4me3. In order to accurately quantify histone methylation and observe differences in modification densities, we performed ICeChIP-seq for H3K4me3 in MV4;11 cells treated with pinometostat. H3K4me3 is deposited at promoters during active transcriptional initiation and promotes gene expression through several established mechanisms (*Vermeulen et al., 2007*; *Yokoyama et al., 2004*; *Krogan et al., 2002*). Surprisingly, pinometostat treatment increased H3K4me3 at transcription start sites (TSS's) genome-wide, with the largest increases at genes downregulated by pinometostat (*Figure 7C*). Pinometostat-downregulated MLL-AF4 targets had the highest H3K4me3 levels of all gene categories examined (*Figure 7C*), not only at the TSS but spreading downstream into the gene body, suggesting that the MLL-fusion protein is driving this increase. Similar upregulation is observed after only 4 days of pinometostat treatment (*Figure 7—figure supplement 1D*), and in two additional ICeChIP-seq experiments at 7 days of treatment we observed increases in promoter H3K4me3 by qPCR consistent with our ICeChIP-seq results at genes up and downregulated by pinometostat as well as MLL-AF4 targets (*Figure 7—figure supplement 1E*). Reductions in H3K79me2 from pinometostat treatment are poorly correlated with increases in H3K4me3 (*Figure 7—figure supplement 1H*), incompatible with a direct antagonism cross-talk mechanism. MLL-AF4 targets, particularly those downregulated by low-dose pinometostat, are slightly skewed toward increases in H3K4me3. Despite gains of the H3K4me3 mark during treatment, these genes are downregulated, consistent with a decoupling of active transcription initiation from productive elongation, the latter of which is more effectively correlated with H3K79me2 and H3K36me3 (*Guenther et al., 2007*). However, when examining H3K4me3 fold changes stratified by gene expression level, we do not observe larger increases in H3K4me3 at more transcriptionally active genes, suggesting that the increase in H3K4me3 is not due to iterative methylation by MLL1 associated with a stalled pol II (*Figure 7—figure supplement 1I*).

Intriguingly, the putative antagonism between modifications is not apparent in global H3K4me3 levels during DOT1L inhibition (*Figure 7—figure supplement 1A*). However, reductions in H3K4me3 from MLL1 inhibitor treatment are also not readily apparent by Western blot, similar to what has been observed in other studies (*Cao et al., 2014*; *Figure 7—figure supplement 1B*). Conversely, global increases in H3K79me2 are more pronounced when treating cells with the MLL1 inhibitors (*Figure 7—figure supplement 1C*). Treatment with the MLL1 inhibitors also reduced STAT5A phosphorylation, suggesting that this orthologous means of disrupting MLL-fusion gene activation also reduces FLT3-ITD/STAT5A signaling (*Figure 7—figure supplement 1C*).

As with the DOT1L and FLT3 inhibitors, overexpression of *STAT5A-CA* was able to partially rescue survival of MV4;11 cells treated with MI-503 (*Figure 7D*), with the degree of rescue corresponding to the amount of *STAT5A* expression in each clone (*Figure 5A*). When treated with the MM-401 inhibitor, STAT5A-CA clone 1 (with the highest exogenous *STAT5A-CA* expression), completely rescued proliferation (*Figure 7E*). Unexpectedly, clones 2 and 3, that express *STAT5A-CA* at lower levels both displayed reduced proliferation when treated with MM-401 compared to WT or GFP-expressing cells (*Figure 7E*).

We observed an additive effect on proliferation when MV4;11 cells were co-treated with the MLL1 and DOT1L inhibitors (*Figure 7—figure supplement 1F*), suggesting that either the inhibitors

target different sets of genes through different mechanisms, or have an additive effect on the same genes. To distinguish between these two models, we compared gene expression of several MLL-AF4 and STAT5A targets in MV4;11 cells treated with MI-503 alone or MI-503 with pinometostat for 7 days (*Figure 7F*). Akin to low dose DOT1L inhibitor treatment, MI-503 reduced expression of *FLT3*, *MEF2C*, *ARID3B* and *PIM1*. The reduction in *FLT3* expression was only $30 \pm 10\%$ but doubled to $60 \pm 10\%$ when both inhibitors were used, recapitulating the 60% reduction observed with pinometostat alone. MI-503 had no significant effect on *PBX3* expression but both inhibitors reduced *PBX3* expression to $50 \pm 10\%$, the same as the DOT1L inhibitor alone. However, unlike low-dose pinometostat, MI-503 treatment starkly reduced expression of *HOXA9* and *MEIS1* and combination treatment further reduced *MEIS1* expression from 40% to 30%. Taken together, low dose MLL1 and DOT1L inhibitors downregulate different, yet partially overlapping sets of genes (with *FLT3*, *MEF2C*, and *PIM1* in common), that are necessary for MLL-rearranged leukemia, consistent with the synergism arising from largely distinct pathways.

We wondered whether the *STAT5A-CA*-mediated rescue of proliferation in MV4;11 cells treated with MI-503 coincided with a rescue of STAT5A target genes. We examined expression of these targets in our 3 *STAT5A-CA* clones after treating cells with MI-503 for 7 days and observed increased expression of the STAT5A target genes *PIM1*, *PIM2*, and *ARID3B* (*Figure 7G*). Collectively, these data suggest that downregulation of *FLT3-ITD*, and crucially, reductions in STAT5A phosphorylation and gene activation are more sensitive to perturbations of MLL-fusion-mediated gene activation and are the main source of inhibitor effects on leukemia cell survival when expression of the canonical MLL-r proliferation mediators *HOXA9* and *MEIS1* are not substantially affected (model, *Figure 7—figure supplement 1G*).

## Discussion

Little is known about why MLL-r leukemia cell lines have such disparate sensitivities to DOT1L inhibitors or how MLL-fusions might cooperate with co-occurring lesions. By investigating the effects of a DOT1L inhibitor at a low, as yet unexplored concentration, we revealed that MLL-r cell lines carrying *FLT3-ITD* lesions are more sensitive to DOT1L inhibition. We observed that a subset of MLL-AF4 targets, including *FLT3*, have aberrantly high H3K79me2 and that low-dose inhibitor treatment downregulates these genes, dramatically depleting H3K79me2, while resulting in increased H3K4me3 at promoters and reduced H3K27me3 genome-wide. Similar effects of both pinometostat and SGC0946, a distinct DOT1L inhibitor, argues that these effects are due to depletion of H3K79me2 rather than off-target effects of pinometostat. Our findings illustrate how MLL-fusions can cooperate mechanistically with *FLT3-ITD* mutations to facilitate leukemogenesis and how PRC2 function may be important for that disease state. FLT3-ITD-mediated STAT5A activation is crucial to the MLL-AF4 expression profile, potentially through direct interaction of STAT5A with HOXA9 and coactivation of some targets such as PIM1.

### The FLT3-ITD signaling pathway accounts for the bulk of low-dose DOT1L inhibitor toxicity

A subset of MLL-AF4 targets were downregulated by low-dose DOT1L inhibition and the FLT3 locus was impacted earlier than other MLL-AF4 targets. *FLT3* expression was downregulated after only 2 days of low-dose pinometostat treatment, coinciding with reduced proliferation, increased apoptosis and gene expression changes consistent with differentiation. Reductions in *FLT3-ITD* expression precede reductions in other MLL-AF4 targets including *PBX3* and *MEF2C*, arguing that these effects are more primary or sensitive to DOT1L function. Although PBX3 interacts with both HOXA9 and MEIS1 to facilitate leukemogenesis and regulate the expression of common targets including *FLT3* (*Li et al., 2016*; *Li et al., 2013b*) we observed that *PBX3* expression could also be reduced by either FLT3 knockdown or inhibition (*Figure 4C*). These results are in agreement with previous findings that *PBX3* was significantly upregulated in *FLT3-ITD*[+] compared to WT *FLT3*, karyotypically normal AML patient samples (*Cauchy et al., 2015*).

The FLT3 receptor has an outsized effect on myeloid differentiation and proliferation through its regulation of several myeloid transcription factors (*Mizuki et al., 2003*; *Rosen et al., 2010*), accounting for its predominance in AML patients (*Nagel et al., 2017*; *Levis and Small, 2003*; *Mizuki et al., 2003*). Although stable transfection of *FLT3-ITD* has been observed to downregulate the PU.1 and

C/EBPα transcription factors and regulators of myeloid differentiation (*Mizuki et al., 2003*), we detected no discernable change in *SPI1* (PU.1) expression and a surprising ~twofold downregulation of *CEBPA* (C/EBPα) in MV4;11 cells (*Figure 1—figure supplement 1F*) treated with low-dose pinometostat. Much of the *FLT3-ITD*-driven effects on proliferation, inhibition of apoptosis and differentiation have been attributed to the activation of STAT5A (*Mizuki et al., 2003*; *Moore et al., 2007*; *Rosen et al., 2010*; *Zhou et al., 2009*; *Spiekermann et al., 2003*). Constitutively active *Stat5a* can render mouse Ba/F3 cells growth factor-independent and resistant to apoptosis through upregulation of the *Pim1-2* protooncogenes (*Adam et al., 2006*; *Kim et al., 2005*; *Santos et al., 2001*). We observe that 100 nM pinometostat downregulates *FLT3-ITD* with concomitant reductions in STAT5A phosphorylation and diminished expression of the STAT5A target genes *PIM1* and *ARID3B,* suggesting that low-dose DOT1L inhibition is able to disrupt FLT3-ITD-mediated signaling and downstream oncogene activation.

Exogenous expression of constitutively active human *STAT5A* (*STAT5A-CA*) in MV4;11 cells treated with 100 nM pinometostat rescues cells from apoptosis, almost completely rescues proliferation, and restores *PIM1* and *ARID3B* gene expression, suggesting that most of the toxicity from low-dose DOT1L inhibition is through loss of STAT5A activation. The ability of ectopic *STAT5A-CA* expression to rescue orthologous perturbations to MLL-fusion-mediated gene activation and proliferation from MLL1 inhibitors suggests that STAT5A activation is necessary for leukemogenesis and maintenance of the proliferative gene expression profile including *PIM1* in this context. Interestingly, *PIM1* is a downstream target of both FLT3-ITD and HOXA9 (*Huang et al., 2012*; *Kim et al., 2005*). Although both factors regulate *PIM1* expression, the FLT3-ITD axis is more sensitive and is responsible for *PIM1* downregulation with low-dose DOT1L inhibitor treatment in MLL-r leukemia also bearing the *FLT3-ITD* mutation. FLT3-ITD-mediated activation of STAT5A may promote HOXA9 localization to the *PIM1* locus or complement it, thereby facilitating expression of this common target and leukemogenesis.

*PIM1* activation by both STAT5A and HOXA9 represents a common coregulation scenario for these hematopoietic transcription factors. Indeed, De Bock et al. discovered that HOXA9 binding sites have significant overlap with STAT5A, PBX3, and C/EBP targets genome-wide (*de Bock et al., 2018*). We observed downregulation of both *PBX3* and *C/EBPA* by 100 nM DOT1L inhibition. It is possible that the dependence of MLL-r, *FLT3-ITD*⁺ leukemia on *FLT3-ITD* expression may be due to HOXA9 requiring STAT5A and/or PBX3 and C/EBPA to cooperatively bind select target genes. Huang et al. found that HOXA9 and MEIS1 preferentially localized to enhancer regions enriched with STAT5 binding motifs (*Huang et al., 2012*) and identified STAT5A and C/EBPA in complex with HOXA9. Furthermore, *HOXA9* knockdown reduced STAT5A binding at common target sites (*Huang et al., 2012*). If HOXA9 depends on STAT5A for chromatin localization then low-dose DOT1L inhibition may reduce HOXA9 binding at enhancer regions, reducing HOXA9 target gene activation without affecting *HOXA9* expression.

In addition to gene activation, STAT5A phosphorylation also results in gene repression, modulating the immune response and differentiation (*Moore et al., 2007*; *Zhu et al., 2003*). Viral transduction of constitutively active *Stat5a* affects T cell differentiation by repressing IFN-γ production (*Zhu et al., 2003*; *Rani and Murphy, 2016*). We found 2007 genes upregulated with 100 nM pinometostat treatment, including many MHC class II genes with large fold-changes that significantly overlapped with a set of genes consistently downregulated in FLT3-ITD+ (KN) leukemia samples (*Cauchy et al., 2015*). Indeed, GO analysis of the pinometostat-upregulated genes indicated significant enrichment for the 'IFN-γ-mediated signaling pathway' and other immune-related categories (*Figure 2—figure supplement 1A*). Despite the increase in expression of IFN-γ-regulated genes we saw barely measurable levels of *IFNG* (IFN-γ) and no increase in expression with pinometostat treatment (*Figure 2—figure supplement 1B*). Many components of the IFN-γ pathway, such as IRF4 and IRF5 are involved in macrophage differentiation, a functional consequence of *DOT1L* deletion or inhibition that has been observed in other studies (*Bernt et al., 2011*; *Daigle et al., 2011*; *Mossadegh-Keller et al., 2013*; *Yamamoto et al., 2011*). With pinometostat treatment we observed upregulation of *CSF1R* and *CSF3R,* targets of IRF4 and critical signaling inducers of macrophage and neutrophil differentiation, respectively (*Figure 2—figure supplement 1.E*; *Mossadegh-Keller et al., 2013*; *Klimiankou et al., 2017*). Additionally, expression increases in the macrophage cell surface markers *ITGAM* (CD11b), *ITGAX* (CD11c), and *CD86* suggest these cells are differentiating to a more macrophage-like state (*Figure 2—figure supplement 1C*), consistent with previous

observations from DOT1L deletion and from a study using the DOT1L inhibitor EPZ004777 (*Bernt et al., 2011*; *Daigle et al., 2011*).

## Extensive histone modification cross-talk contributes to the survival of MLL-r, FLT3-ITD[+] leukemia

*FLT3* is part of a subset of MLL-AF4 targets that are more sensitive to reductions in H3K79me2 than even the *HOXA9* and *MEIS1* oncogenes. We observed that MLL-AF4 targets (*Kerry et al., 2017*) that are downregulated by 100 nM pinometostat have higher levels of H3K79me2 than even the most highly expressed genes and show the largest reductions in methylation when treated with pinometostat. (*Figure 3B*). The greater reductions in H3K79me2 levels at downregulated genes is likely a contributing factor to their loss of gene expression. H3K79me2 hypermethylation antagonizes SIRT1 localization to MLL-AF4 targets, preventing H3K9ac and H3K16ac deacetylation, thereby facilitating gene expression (*Chen et al., 2015a*). However, there are stark differences in methylation density and susceptibility to DOT1L inhibition even among MLL-fusion targets. MLL-AF4 'spreading' genes (*Kerry et al., 2017*) had H3K79me2 levels comparable to those MLL-AF4 targets whose expression was downregulated by pinometostat. Yet only 31% of 'spreading genes' were downregulated by 100 nM pinometostat, suggesting that effects on gene expression from depletion of H3K79me2 could be governed by other factors including changes to the distribution of other chromatin modifications.

To our surprise, the pinometostat-induced activation of MHC class II genes we observed did not appear to result from a loss of H3K27me3-mediated repression, despite PRC2 subunit downregulation. Treatment with PRC2 inhibitor EI1 had no effect on *CIITA* or MHC class II gene expression but significantly reduced proliferation in MV4;11 cells (*Figure 6E* and *Figure 6—figure supplement 1B*). A growing body of evidence supports an essential role for the PRC2 complex in MLL-r leukemogenesis– PRC2 is necessary for *MLL-AF9*-induced leukemogenesis in mouse progenitor cells and cooperates with MLL-AF9 to promote self-renewal of acute myeloid leukemia cells (*Shi et al., 2013*; *Neff et al., 2012*). The observed downregulation of the MLL-AF4 target oncogenes upon EZH2 inhibition (*Figure 6F*), suggests that MLL-fusion-mediated gene activation is in some way dependent on PRC2 methyltransferase activity. Consistent with this idea, ectopic expression of *EZH2* was able to provide a small but significant proliferation rescue when treating cells with 100 nM pinometostat (*Figure 6G*).

We identified pinometostat-induced increases in H3K4me3 at promoters genome-wide (*Figure 7C*). Although H3K4me3 promotes transcriptional initiation (*Vermeulen et al., 2007*; *Chang et al., 2010b*), the largest H3K4me3 increases were at downregulated MLL-AF4 targets that had the largest decreases in H3K79me2. Although DOT1L inhibition reduces global H3K27me3, this is unlikely to explain the massive increases in H3K4me3 that we observe (*Kim et al., 2013*; *Hanson et al., 1999*). Studies in human embryonic stem cells and mouse preadipocytes observed no genome-wide increases in H3K4me3 upon *EZH2* knockout and reductions in H3K27me3 (*Collinson et al., 2016*; *Wang et al., 2010*). The increase in H3K4me3 does not appear to be the result of a stalled transcriptional complex containing MLL1 near the TSS as more highly expressed genes do not show greater fold-changes in H3K4me3 upon pinometostat treatment (*Figure 7—figure supplement 1I*). Additionally, the absence of an anti-correlation between H3K79me2 loss and increases in H3K4me3 suggests that there is not a direct antagonism between these modifications at genes (*Figure 7—figure supplement 1H*). The increase in H3K4me3 further into the gene body of 'MLL-spreading genes' and the strong skew of downregulated MLL-AF4 targets toward increases in H3K4me3 suggests that the buildup of this modification is possibly the result of reduced recruitment or activity of the H3K4me2-histone demethylase LSD1. Previous studies have observed that knockout or inhibition of LSD1, a component of the MLL-supercomplex, results in apoptosis and differentiation of MLL-r cells, inhibits leukemogenesis in mouse models and increases H3K4me2/3 at MLL-target genes (*Harris et al., 2012*; *Feng et al., 2016*; *McGrath et al., 2016*; *Fang et al., 2017*). A more localized antagonism could be potentially mediated through H3K79me2-mediated recruitment/activation of LSD1.

### Broader clinical implications

In light of the heightened sensitivity we observed for a non-MLL-rearranged FLT3-ITD leukemia cell line to DOT1L inhibition and the coinciding reductions in cell viability and FLT3 expression, small molecules such as pinometostat may prove effective in treating the 30–40% of AML bearing FLT3-ITD mutations. Although several FLT3 inhibitors have undergone clinical trials, drug resistance has emerged as a formidable and so far, insurmountable barrier to an effective treatment. A previous study observed that siRNAs targeting FLT3 expression increased the efficacy of the FLT3 inhibitor tandutinib (*Walters et al., 2005*). As a way of circumventing the difficulties associated with thera-peutic siRNA delivery, DOT1L inhibitors that reduce FLT3 expression might serve as an effective adjunct treatment with FLT3 inhibitors. Our mechanistic studies provide impetus for exploration of these ideas in pre-clinical or patient-derived FLT3-ITD or MLL-PTD leukemias.

## Materials and methods

### Cell culture

Human MV4;11 and MOLM13 leukemia cells and MLL1 inhibitor MM-401 were gifts from the labora-tory of Yali Dou at the University of Michigan. MV4;11, MOLM13, THP-1, and K562 cells were vali-dated by STR profiling through ATCC. Human THP-1 leukemia cells (cat # TIB-202) were purchased from American Type Culture Collection (ATCC). Human SEM (ACC546), EOL-1 (ACC386), and PL-21 (ACC536) leukemia cells were obtained from DSMZ- the German Collection of Microorganisms and Cell Cultures GmbH. Experiments using these purchased cell lines were performed within 1 year of receipt of the cell lines. All cell lines tested negative for mycoplasma with the Universal Mycoplasma Detection Kit from ATCC (cat # 30–1012K). Cells were cultured in RPMI-1640 medium containing 10% (v/v) FBessence (Seradigm cat # 3100–500), 1% L-glutamine at 37°C in humidified air containing 5% $CO_2$. DOT1L inhibitor pinometostat (EPZ5676, Cayman Chemical cat # 16175), EZH2 inhibitor EI1 (Cayman Chemical cat # 19146–1), FLT3 inhibitor tandutinib (MLN518) (Selleckchem cat # S1043), MI-503 (Selleckchem cat # S7817), and PIM1 inhibitor (MedChemExpress cat # HY-15604) were resuspended in DMSO. Doxycycline (Alfa Aesar cat # J60422) was resuspended in water.

Plasmid generation pCMV-Gag-Pol plasmid, encoding HIV-1 derived *gag*, and *pol*, the pCMV-VSV-G vector encoding VSV-G envelope gene, pTRIPZ-YFP-EED and Tet-pLKO were purchased from Addgene. pTRIPZ-STAT5a-CA and pTRIPZ-FLT3-ITD were created by cloning *STAT5A* and *FLT3-ITD* from cDNA from MV4;11 cells. STAT5A-CA mutations were introduced at H298R and S710F and genes were inserted into the pTRIPZ plasmid at restriction sites AgeI and MluI. shRNA constructs were created by inserting annealed oligos of shRNA sequences (*Supplementary file 1*) purchased from IDT into Tet-pLKO at the AgeI and EcoRI restriction sites.

### RNA-seq and gene expression analysis

Exponentially growing MV4;11 cells were grown in 150 mm² tissue culture-treated plates (Corning cat # 0877224) in 30 ml media ± 100 nM pinometostat for 7 days. Every 2 days, cells were spun down at 500 x g 5 min then resuspended in media ± 100 nM pinometostat. On day 7, $1 \times 10^7$ cells were spun down at 500 x g 5 min then cells were resuspended in 1 ml Trizol reagent (Life Technologies cat# 15596018), incubated 5 min at 25°C then 200 µl chloroform was added and samples were shaken rigorously for 15 s then incubated 3 min at 25°C and spun down 12,000 x g 15 min at 4°C. The aqueous layer (~ 500 µl) was removed and mixed with 500 µl EtOH and added to a Zymo Research RNA Clean and Concentrator column (cat # 11-353B) and spun 12,000 x g 1 min. A total of 100 µl DNase I (1:10 in buffered dH20) (Thermo Fisher Scientific cat # en0521) was added to the col-umn and then spun 500 x g 5 min, incubated 15 min at 25°C and then spin 12,000 x g for 30 s. Com-bined 200 µl RNA binding buffer with 300 µl EtOH and then spun 12,000 x g for 30 s and the flow through was discarded. After each of the following were added to the column, it was spun down 12,000 x g for 30 s and the flow through was discarded: 400 µl RNA prep buffer; 700 µl RNA wash buffer; and 400 µl RNA wash buffer. RNA was eluted from column with 30 µl RNase-free dH₂O. Added RNA standards to 2 µg of each RNA sample- Add the equivalent of 10 copies/cell yeast RAD51; 30 copies/cell RNL2; 200 copies/cell E coli MBP; and 2000 copies/cell yeast SUMO to each sample then proceed with rRNA removal Ribo Zero Gold kit (Illumina cat # MRZ11124C) according to manufacturer's protocol. Libraries were prepared using the NEBNext Ultra Directional RNA

Library prep kit (NEB cat # E7420S). Libraries were then sequenced on the Illumina NextSEQ500. Reads were aligned to the hg38 genome assembly using HISAT2 (*Kim et al., 2015*) and differential gene expression analysis was done with Cufflinks (*Trapnell et al., 2012*).

## Reverse transcription and quantitative real-time PCR

RNA was extracted from $10^6$ cells using 500 µl Trizol and following the manufacturer's protocol. One µg RNA was used for reverse transcription with 0.5 µl MMLV HP reverse transcriptase (Lucigen cat # RT80125K) per 20 µl rxn. RNA was then degraded by alkaline hydrolysis by adding 40 µl 150 mM KOH, 20 mM tris base and heating 95℃ 10 min then cooling on ice and quenching with 40 µl 150 mM HCl and then adding 100 µl TE. Gene expression was assayed by real-time PCR in 10 µl reactions with 0.5 µl cDNA and 5 µl PowerUP SYBR Green master mix (Applied Biosystems cat # A25742) per reactions. qPCR was run on the Bio-Rad thermocycler CFX96 or CFX384 using the program: 50℃ 2:00, 95℃ 2:00, then 40 cycles 95℃ 0:15, then 60℃ 1:00. Expression was normalized to 18S rRNA. Primer sets are listed in *Supplementary file 1*.

## Cell proliferation assay

Cells were seeded at $10^5$ cells/ml in 80 µl in clear bottom 96-well plates (Corning 07200566) in three replicates. Everyday 40 µl of culture was transferred to 40 µl media in a new plate. On odd days, 30 µl of Cell TiterGlo 2 (Promega cat # G924A) was added to the remaining 40 µl culture and incubated 10 min at room temperature on a shaker at 600 rpm. Luminescence was measured on a Tecan Infinite F200 Pro plate reader and fraction viability was determined from the luminescence of treated over untreated cells.

## Apoptosis assay

Exponentially growing cells were incubated with increasing concentrations of pinometostat for 7 days in 3 ml media in six-well plates in three experimental replicates. $10^6$ were harvested from each plate and washed twice in 1 ml PBS then resuspended in 1 ml binding buffer as per BD Biosciences manufacturer's protocol. Add 5 µl FITC-conjugated Annexin V (BD Biosciences cat# 556420) and 2 µl propidium iodide (Alfa Aesar cat # J66584) to 100 µl cells and incubate 15 min at 25℃ in the dark. Cells were then sorted on the BD FACSAriaII device for propidium iodide or FITC (Annexin V) positive cells. Data was analyzed using FlowJo software (Tree Star).

## Calibrated chromatin immunoprecipitation sequencing (ICeChIP-seq)

Native, internally calibrated ICeChIP-seq was carried out as described previously for H3K4me3 and H3K27me3 (*Grzybowski et al., 2015*; *Grzybowski et al., 2019*). A modified protocol was used for H3K79me2 that included cross-linking and denaturation because of greater difficulty in immunoprecipitation of this modification, likely due to reduced accessibility of this mark within the more highly structured nucleosome core. Briefly, MV4;11 cells were exposed to a gentle detergent lysis and spun through a sucrose gradient to obtain nuclei from 20 million cells. Nucleosome standards were added and then the chromatin was digested with micrococcal nuclease and purified using hydroxy apatite (HAP) resin. Of 280 µl total HAP-purified chromatin, 150 µl was removed for denaturative ICeChIP and crosslinked in 0.25% formaldehyde for 8 min on a nutator at 25℃, then quenched by adding 1M Tris pH 7.5 to 200 mM and incubated 5 min at 25℃ on a nutator. Fifty µl of cross-linked chromatin was used for denaturation and 2.5 µl 20% SDS was added to 1% SDS final concentration and the sample was incubated 1 min at 55℃, then immediately placed on ice. This was then diluted with nine volumes water (450 µl) and 100 µl was used for each IP. For native ICeChIP, the HAP-purified chromatin was diluted to 20 µg/ml and the indicated amounts of chromatin, adjusted for the relative approximate abundance of each modification were added to the antibody/beads for immunoprecipitation. Antibodies for both the DMSO- and pinometostat-treated samples were processed together (12 µl antibody-bound beads per IP). Three µg of anti-H3K79me2 (Abcam cat # ab3594, lot # GR173874); 3 µg of anti-H3K4me3 (Abcam cat # 12209, lot # GR275790-1); and 0.6 µg of anti-H3K27me3 (Cell Signaling cat # 9733, lot # 8) were used per IP. For crosslinked IPs include 1 hr 65℃ after proteinase K digest. Libraries were prepared using the NEBNext Ultra II DNA Library prep kit (NEB cat # E7645). Three cycles of PCR amplification were used for native inputs, four cycles for denatured inputs and H3K27me3 IPs, seven cycles for H3K4me3 IPs and 10 cycles

for H3K79me2 IPs. Analysis of histone methylation density (HMD) was carried out using the scripts and workflow from *Grzybowski et al., 2019*.

## Western blotting

Ten Vl whole cell extracts of $2 \times 10^5$ cells in 40 µl 6X SDS loading buffer were run on 4–14% bis-tris gel (Invitrogen cat # NP0335). Membranes were transferred by semi-dry apparatus (Bio-Rad Transblot cat # 170–3940) at 200 mA, 25 V for 35 min to 0.45 µm nitrocellulose membrane (Millipore cat # IPVH00010). Membranes were then blocked for 1 hr with TBS-T 1% ECL Prime blocking reagent (GE Healthcare cat # RPN418) at 25°C on an orbital shaker and blotted with primary antibody for 1 hr at 25°C with gentle agitation. Membranes were then washed three times for 5 min while shaking with TBS-T and then incubated with secondary antibody at 25°C for 1 hr while shaking. A complete list of antibodies can be found in *Supplementary file 2*.

## Transfection for lentiviral particle generation

Lentiviral particles were produced by Fugene transfection of the 293T packaging cell line in a six-well plate at ~70% confluency with pCMV-Gag-Pol, pCMV-VSV-G and 2 µg of the plasmid encoding the gene or shRNA of interest using a 3:1:4 ratio, respectively. Lentiviral particle enriched supernatants were collected 72 hr after transfection for immediate transduction.

## Lentiviral transduction

Four x $10^5$ MV4;11 cells suspended in 1 ml RPMI-1640 medium containing 10% FBessence in a six-well plate were transduced by adding 2.5 ml of 0.45 µm filtered viral supernatants from 293 T cells. Then 0.8 µl polybrene (EMD Millipore cat. # TR-1003-G)/ml transduction reagent was added to the media and the plates were wrapped with parafilm and spun down at 2000 rpm for 2 hr at room temperature then incubated O/N at 37°C in humidified air containing 5% $CO_2$. After 12 hr, cells were spun down and resuspended in RPMI-1640 10% FBessence. After 24 hr, 0.5 µg/ml puromycin was added to the wells and this selection media was refreshed every 3 days to select for transduced cells. Individual clones were purified by diluting cell cultures to 1 cell/100 µl and then plating 100 µl aliquots in a 96-well plate. Wells were visually assessed for individual clones and then grown out.

## Acknowledgements

This work was supported by the American Cancer Society (130230-RSG-16-248-01-DMC to AJR) and National Institutes of Health (R01-GM115945 to AJR). WFR was supported by NIH Molecular and Cell Biology training grant T32-GM007183. We thank Peter Faber and Mikayla Marchuk of the University of Chicago Functional Genomics Core and Balaji Manicassamy for his kind help with lentivirus protocols. We also thank Mary Beth Neilly, Elizabeth Davis and the le Beau lab for their help in securing leukemia cell lines for our experiments.

## Additional information

### Funding

| Funder | Grant reference number | Author |
| --- | --- | --- |
| American Cancer Society | 130230-RSG-16-248-01-DMC | Alexander J Ruthenburg |
| National Institutes of Health | R01-GM115945 | Alexander J Ruthenburg |
| NIH | T32-GM007183 | William F Richter |

The funders had no role in study design, data collection and interpretation, or the decision to submit the work for publication.

### Author contributions

William F Richter, WFR conceived of and designed the study. Nearly all experiments were conducted by WFR under supervision from AJR, including ICeChIP-seq for all datasets except AR15. WFR also

conducted bioinformatic analyses; Rohan N Shah, RNS developed the ICeChIP methodology used in this study and performed ICeChIP-seq on the AR15 dataset. RNS also conducted bioinformatic analyses; Alexander J Ruthenburg, AJR conceived of and designed this study, and supervised experiments conducted by WFR. Funding acquisition was conducted by AJR

## Author ORCIDs
William F Richter  https://orcid.org/0000-0003-4469-6428
Alexander J Ruthenburg  https://orcid.org/0000-0003-2709-4564

## Decision letter and Author response
Decision letter https://doi.org/10.7554/eLife.64960.sa1
Author response https://doi.org/10.7554/eLife.64960.sa2

# Additional files
## Supplementary files
- Source data 1. Blotting source files.
- Source data 2. Blotting source files.
- Source data 3. Blotting source files.
- Supplementary file 1. Oligonucleotide sequences.
- Supplementary file 2. Antibody information.
- Transparent reporting form

## Data availability
ICeChIP-seq and RNA-seq data have been deposited in GEO under the accession code GSE162441.

The following datasets were generated:

| Author(s) | Year | Dataset title | Dataset URL | Database and Identifier |
|---|---|---|---|---|
| Richter WF | 2020 | RNA-seq analysis of 10 nM pinometostat vs. DMSO-treated MV4;11 cells | https://www.ncbi.nlm.nih.gov/geo/query/acc.cgi?acc=GSM4952087 | NCBI Gene Expression Omnibus, GSM4952087 |
| Richter WF, Shah RN | 2020 | ICeChIP-seq of H3K79me2, H3K4me3, H3K27me3, H3K36me3 from MV4;11 cells treated with 10 nM pinometostat or DMSO | https://www.ncbi.nlm.nih.gov/geo/query/acc.cgi?acc=GSM4952104 | NCBI Gene Expression Omnibus, GSM4952104 |

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

# Appendix 1

**Appendix 1—key resources table**

| Reagent type (species) or resource | Designation | Source or reference | Identifiers | Additional information |
|---|---|---|---|---|
| cell line (*Homo-sapiens*) | SEM | DSMZ | ACC546 | |
| cell line (*Homo-sapiens*) | MOLM13 | Duo laboratory University of Michigan | | Verified by STR profiling ATCC |
| cell line (*Homo-sapiens*) | THP-1 | ATCC | TIB-202 | Verified by STR profiling ATCC |
| cell line (*Homo- sapiens*) | MV4;11 | Duo laboratory University of Michigan | | Verified by STR profiling ATCC |
| Gene (*Homo- sapiens*) | STAT5A-CA | This paper | | Constitutively active human STAT5A. |
| recombinant DNA reagent | pTRIPZ-STAT5A-CA (plasmid) | This paper | | Lentiviral construct to transfect and express the gene. |
| recombinant DNA reagent | pTRIPZ-EZH2 (plasmid) | This paper | | Lentiviral construct to transfect and express the gene. |
| recombinant DNA reagent | Tet-pLKO-puro | Addgene | Cat # 21915 | |
| recombinant DNA reagent | PLKO-shFLT3 (plasmid) | This paper | | Lentiviral construct to transfect and knock down gene. |
| recombinant DNA reagent | PLKO-shscrambled (plasmid) | This paper | | Lentiviral construct to transfect and knock down gene. |
| recombinant DNA reagent | pTRIPZ-GFP (plasmid) | This paper | | Lentiviral construct to transfect and express the gene. |
| antibody | anti-MEIS1 (Rabbit polyclonal) | EMD Millipore | Cat# ABE2864 | WB (1:1000) |
| antibody | anti-H3 (Rabbit polyclonal) | Active Motif | Cat#: 61277 | WB (1:5000) |
| antibody | anti-H3K79me2 (Rabbit polyclonal) | Abcam | Cat# ab3594 | WB (1:2000) |
| antibody | anti-H3K27me3 (Rabbit polyclonal) | Cell Signaling | Cat# 9733S | WB (1:1000) |
| antibody | anti-H3K4me3 (mouse monoclonal) | Cell Signaling | Cat# 9751S | WB (1:1000) |
| antibody | anti-H4 (Rabbit polyclonal) | Active Motif | Cat# 61299 | WB (1:1000) |
| antibody | anti-H2B (Rabbit polyclonal) | Proteintech | Cat# 2S2899-2 | WB (1:5000) |
| antibody | anti-FLT3 (Rabbit polyclonal) | Cell Signaling | Cat. #: 33462S | WB (1:500) |
| antibody | anti-GAPDH (Rabbit polyclonal) | Cell Signaling | Cat# 5174S | WB (1:5000) |
| antibody | anti-STAT5A (Rabbit polyclonal) | Cell Signaling | Cat# 94205T | WB (1:1000) |

*Continued on next page*

*Appendix 1—key resources table continued*

| Reagent type (species) or resource | Designation | Source or reference | Identifiers | Additional information |
|---|---|---|---|---|
| antibody | anti-p- STAT5 (Rabbit polyclonal) | Cell Signaling | Cat# 9359S | WB (1:1000) |
| antibody | anti-EZH2 (Rabbit polyclonal) | Cell Signaling | Cat. #: 5246S | WB (1:1000) |
| antibody | anti-LEDGF (Rabbit polyclonal) | Bethyl | Cat# A300-848A | WB (1:2000) |
| antibody | anti-RBBP5 (Rabbit polyclonal) | Bethyl | Cat# A300-109A | WB (1:1000) |
| antibody | anti-HNRNPK (Rabbit polyclonal) | Abcam | Cat# ab70492 | WB (1:5000) |
| antibody | anti-MBD3 (Rabbit polyclonal) | Bethyl | Cat. #: A302-529A | WB (1:1000) |
| antibody | anti-rabbit (secondary HRP-conjugated) | Cell Signaling | Cat# 7074S | WB (1:10000) |
| antibody | anti-mouse (secondary HRP-conjugated) | Bethyl | Cat# 31432 | WB (1:10000) |
| chemical compound, drug | Pinometostat (EPZ5676) | Cayman Chemical | Cat# a16175 | |
| chemical compound, drug | Tandutinib (MLN518) | Selleckchem | Cat. #: S1043 | |
| chemical compound, drug | EI1 | Cayman Chemical | Cat# 19146–1 | |
| chemical compound, drug | MI-503 | Selleckchem | Cat. #: S7817 | |
| chemical compound, drug | PIM1 inhibitor | MedChemExpress | Cat# HY-15604 | |
| chemical compound, drug | MM-401 | Duo laboratory University of Michigan | | |
| chemical compound, drug | Propidium iodide | Alfa Aesar | Cat # J66584 | |
| recombinant DNA reagent | pCMV-Gag-Pol (plasmid) | Manicassamy lab | | Lentiviral construct to transfect for viral particle assembly. |
| recombinant DNA reagent | pCMV-vsvg (plasmid) | Manicassamy lab | | Lentiviral construct to transfect for viral particle assembly. |
| recombinant DNA reagent | pTRIPZ-FLT3-ITD | This paper | | Lentiviral construct to transfect and express the gene. |
| recombinant DNA reagent | RiboZero Gold | Illumina | Cat # MRZ11124C | Ribosomal rRNA removal from total RNA. |
| commercial assay or kit | NEBNext Ultra Directional RNA Library prep kit | NEB | Cat # E7420S | cDNA library kit. |
| commercial assay or kit | Trizol reagent | Life Technologies | Cat # 15596018 | RNA extraction. |
| commercial assay or kit | Zymo Research RNA Clean and Concentrator | Zymo Research | Cat # 11-353B | |

*Continued on next page*

*Appendix 1—key resources table continued*

| Reagent type (species) or resource | Designation | Source or reference | Identifiers | Additional information |
|---|---|---|---|---|
| commercial assay or kit | NEBNext Ultra II DNA Library prep kit | NEB | Cat # E7645 | DNA library kit. |
| commercial assay or kit | PowerUP SYBR Green master mix | Applied Biosystems | Cat # A25742 | qPCR reagent |
| commercial assay or kit | MMLV HP reverse transcriptase | Lucigen | Cat # RT80125K | |
| commercial assay or kit | Cell TiterGlo 2 | Promega | Cat # G924A | Cell proliferation assay. |
| software, algorithm | HISAT2 | *Kim et al., 2015* | | |
| software, algorithm | Cufflinks | *Trapnell et al., 2012* | | |
| peptide, recombinant protein | FITC-conjugated Annexin V | BD Biosciences | Cat # 556420 | Apoptosis detection reagent. |
| sequence-based reagent | HLD-DRA qPCR F | This paper | PCR primers | CTCAGGAATC ATGGGCTATCAA |
| sequence-based reagent | HLA-DRA qPCR R | This paper | PCR primers | CTCATCACCAT CAAAGTCAAACAT |
| sequence-based reagent | HLA-DRB1 qPCR F | This paper | PCR primers | GTGACACTGAT GGTGCTGAG |
| sequence-based reagent | HLA-DRB1 qPCR R | This paper | PCR primers | GCTCCGTCCC ATTGAAGAAA |
| sequence-based reagent | MEF2C qPCR F | This paper | PCR primers | GTCTGAGGAC AAGTACAGGAAAA |
| sequence-based reagent | MEF2C qPCR R | This paper | PCR primers | GAGACTGGC ATCTCGAAGTT |
| sequence-based reagent | FLT3 qPCR F | This paper | PCR primers | ATCATATCCCAT GGTATCAGAATCC |
| sequence-based reagent | FLT3 qPCR R | This paper | PCR primers | GAAGCAGATAC ATCCACTTCCA |
| sequence-based reagent | ARID3B qPCR F | This paper | PCR primers | AGACCATACCA AAGATGCTTCC |
| sequence-based reagent | ARID3B qPCR R | This paper | PCR primers | ATCATCACTCC AGGCCAAAC |
| sequence-based reagent | STAT5A qPCR F | This paper | PCR primers | CAGATGCAGG TGCTGTACG |
| sequence-based reagent | STAT5A qPCR R | This paper | PCR primers | TGTCCAAGTC AATGGCATCC |
| sequence-based reagent | PIM2 qPCR F | This paper | PCR primers | ATGTTGACCAA GCCTCTACA |
| sequence-based reagent | PIM2 qPCR R | This paper | PCR primers | TCGATACTCGG CCTCGAA |
| sequence-based reagent | MEIS1 qPCR F | This paper | PCR primers | AGACGATAGAG AAGGAGGATCAA |
| sequence-based reagent | MEIS1 qPCR R | This paper | PCR primers | CCGTGTCATC ATGATCTCTGTT |
| sequence-based reagent | HOXA9 qPCR F | This paper | PCR primers | AGGCGCCTT CTCTGAAA |
| sequence-based reagent | HOXA9 qPCR R | This paper | PCR primers | GTTGGCTGCT GGGTTATTG |

*Continued on next page*

*Appendix 1—key resources table continued*

| Reagent type (species) or resource | Designation | Source or reference | Identifiers | Additional information |
|---|---|---|---|---|
| sequence-based reagent | PBX3 qPCR F | This paper | PCR primers | CCACCAGAT CATGACCATCAC |
| sequence-based reagent | PBX3 qPCR R | This paper | PCR primers | AAGAGCGCTG GTTTCATTCT |
| sequence-based reagent | CEBPA qPCR F | This paper | PCR primers | CCTTCAACG ACGAGTTCCT |
| sequence-based reagent | CEBPA qPCR R | This paper | PCR primers | GCCCGGGT AGTCAAAGTC |
| sequence-based reagent | CSF1R qPCR F | This paper | PCR primers | GCCATCCACCT CTATGTCAAA |
| sequence-based reagent | CSF1R qPCR R | This paper | PCR primers | AGCAGACAG GGCAGTAGT |
| sequence-based reagent | B2M qPCR F | This paper | PCR primers | CTCTCTCTTT CTGGCCTGGAG |
| sequence-based reagent | B2M qPCR R | This paper | PCR primers | TCTGCTGGAT GACGTGAGTA |
| sequence-based reagent | SPI1 qPCR F | This paper | PCR primers | TGCCCTATGA CACGGATCTA |
| sequence-based reagent | SPI1 qPCR R | This paper | PCR primers | GTCCCAGTAAT GGTCGCTATG |
| sequence-based reagent | CSF3R qPCR F | This paper | PCR primers | CTATGGCAAGG CTGGGAAA |
| sequence-based reagent | CSF3R qPCR R | This paper | PCR primers | GGGCTGAGAC ACTGATGTG |
| sequence-based reagent | PIM1 qPCR F | This paper | PCR primers | GTGGAGAAGGA CCGGATTTC |
| sequence-based reagent | PIM1 qPCR R | This paper | PCR primers | TTCTTCAGCAG GACCACTTC |
| sequence-based reagent | PBX3 qPCR F | This paper | PCR primers | CAAAGAAACAT GCCCTGAACTG |
| sequence-based reagent | PBX3 qPCR R | This paper | PCR primers | CTCTGATGCT GAGACCTGTTT |
| sequence-based reagent | 18S (RNA18S5) F | This paper | PCR primers | CGCAGCTAGGA ATAATGGAATAGG |
| sequence-based reagent | 18S (RNA18S5) R | This paper | PCR primers | GCCTCAGTTCC GAAAACCAA |
| sequence-based reagent | CIITA qPCR F | This paper | PCR primers | CTGTGCCTCT ACCACTTCTATG |
| sequence-based reagent | CIITA qPCR R | This paper | PCR primers | GTCGCAGTTGA TGGTGTCT |
| sequence-based reagent | EZH2 qPCR F | This paper | PCR primers | GGAGGATCACCGA GATGATAAAG |
| sequence-based reagent | EZH2 qPCR R | This paper | PCR primers | TTCTGCTGTG CCCTTATCTG |
| sequence-based reagent | EED qPCR F | This paper | PCR primers | CTGGCACAGT AAAGAAGGAGAT |
| sequence-based reagent | EED qPCR R | This paper | PCR primers | GCATCAGCAT CCACGTAAGA |
| sequence-based reagent | MEF2C promoter qPCR F | This paper | PCR primers | TCTGGACGAG TCTGGTTACTT |

*Continued on next page*

*Appendix 1—key resources table continued*

| Reagent type (species) or resource | Designation | Source or reference | Identifiers | Additional information |
|---|---|---|---|---|
| sequence-based reagent | MEF2C promoter qPCR R | This paper | PCR primers | AGGAAGAAGG AGGAGGAAGAG |
| sequence-based reagent | PIM1 promoter qPCR F | This paper | PCR primers | ctcagcgaaa cggagagc |
| sequence-based reagent | PIM1 promoter qPCR R | This paper | PCR primers | cgtatcgattca aacccaaacaa |
| sequence-based reagent | FLT3 promoter qPCR F | This paper | PCR primers | ctttctcaggg cctcaaagat |
| sequence-based reagent | FLT3 promoter qPCR R | This paper | PCR primers | ccgaactctgt cgtttggat |
| sequence-based reagent | ARID3B promoter qPCR F | This paper | PCR primers | acgagaacctc tgaggaaga |
| sequence-based reagent | ARID3B promoter qPCR R | This paper | PCR primers | gctgggaggaaa gtaactaaaga |
| sequence-based reagent | CSF3R promoter qPCR F | This paper | PCR primers | GCAGAACCATT GTGGGTAAAC |
| sequence-based reagent | CSF3R promoter qPCR R | This paper | PCR primers | ggcagatggag aaacaggaa |
| sequence-based reagent | BCL6 promoter qPCR F | This paper | PCR primers | agctcgatctg ctgagtttatg |
| sequence-based reagent | BCL6 promoter qPCR R | This paper | PCR primers | gcctctggaa ttctgagaactaat |
| sequence-based reagent | HOXA9 promoter qPCR F | This paper | PCR primers | GCCTTATGGC ATTAAACCTGAAC |
| sequence-based reagent | HOXA9 promoter qPCR R | This paper | PCR primers | GAGGAGAACC ACAAGCATAGTC |
| sequence-based reagent | MEIS1 promoter qPCR F | This paper | PCR primers | GGAGAGAGA GGGAGAGAAAGAA |
| sequence-based reagent | MEIS1 promoter qPCR R | This paper | PCR primers | CAAATGCAC AAAGCCCTAGC |
| sequence-based reagent | HLA-DRA promoter qPCR F | This paper | PCR primers | CAGAGCGCC CAAGAAGAA |
| sequence-based reagent | HLA-DRA promoter qPCR R | This paper | PCR primers | cctcagcacctac CTTTGATAG |
| sequence-based reagent | intergenic qPCR F | This paper | PCR primers | TACACGACAG AGGACTGGAA |
| sequence-based reagent | intergenic qPCR R | This paper | PCR primers | CCTTCATGGGT GAGGGTAATG |
| sequence-based reagent | scrambled shRNA F | *Yuan et al., 2009* | shRNA construct for gene knockdown | CCGG TTCTCCGAAC GTGTCACGTTT CTCGAG AAACGTGACA CGTTCGGAGAA TTTTT |
| sequence-based reagent | scrambled shRNA R | *Yuan et al., 2009* | shRNA construct for gene knockdown | AATTAAAAA TTCTCCGAA CGTGTCACGTTT CTCGAG AAACGTGACAC GTTCGGAGAA |

*Continued on next page*

*Appendix 1—key resources table continued*

| Reagent type (species) or resource | Designation | Source or reference | Identifiers | Additional information |
|---|---|---|---|---|
| sequence-based reagent | FLT3 shRNA F | *Green et al., 2015* | shRNA construct for gene knockdown | CCGG GCATCCCAGTC AATCAGCTTT CTCGAG AAAGCTGATT GACTGGGATGC TTTTT |
| sequence-based reagent | FLT3 shRNA R | *Green et al., 2015* | shRNA construct for gene knockdown | AATTAAAAA GCATCCCA GTCAATCAGCTTT CTCGAG AAAGCTGAT TGACTGGGATGC |
| sequence-based reagent | GFP shRNA F | *Scheeren et al., 2005* | shRNA construct for gene knockdown | CCGG GCAAGCTGAC CCTGAAGTTCAT CTCGAG ATGAACTTCAGG GTCAGCTTGC TTTTT |
| sequence-based reagent | GFP shRNA R | *Scheeren et al., 2005* | shRNA construct for gene knockdown | AATTAAAAA GCAAGCTGACCC TGAAGTTCAT CTCGAG ATGAACTTCAGG GTCAGCTTGC |

