## [Decision Letter]

**Acceptance summary:**

Your study of non-canonical H3K79 methylation dependent pathways reveals alternative mechanisms to the canonical DOT1L-H3K79me2-HOXA9/MEIS1 axis in MLL-r leukemia, providing novel insight into epigenetic regulation and transcriptional controls as well as potentially new approaches for leukemia therapy. This well executed study will be of interest to the readership of *eLife*.

**Decision letter after peer review:**

Thank you for submitting your article "Non-canonical H3K79me2-dependent pathways promote the survival of MLL-rearranged leukemia" for consideration by *eLife*. Your article has been reviewed by 3 peer reviewers, including Xiaobing Shi as the Reviewing Editor and Reviewer #1, and the evaluation has been overseen by Maureen Murphy as the Senior Editor.

The reviewers have discussed the reviews with one another and the Reviewing Editor has drafted this decision to help you prepare a revised submission.

We would like to draw your attention to changes in our policy on revisions we have made in response to COVID-19 (https://elifesciences.org/articles/57162). Specifically, when editors judge that a submitted work as a whole belongs in eLife but that some conclusions require a modest amount of additional new data, as they do with your paper, we are asking that the manuscript be revised to either limit claims to those supported by data in hand, or to explicitly state that the relevant conclusions require additional supporting data. Our expectation is that the authors will eventually carry out the additional experiments and report on how they affect the relevant conclusions either in a preprint on bioRxiv or medRxiv, or if appropriate, as a Research Advance in eLife, either of which would be linked to the original paper.

The manuscript by Richter et al., describes the discovery of non-canonical pathways dysregulated by low-dose pinometostat, a DOT1L specific inhibitor. MLL rearranged leukemia has been a focus for rational development of small molecule inhibitors targeting key regulatory pathways. Among these efforts, DOT1L inhibitor shows some efficacy in the MLL models. The prevalent view is that DOT1L inhibitors mainly target the HOXA9/MEIS1 pathway and down regulate expression of these leukemia transcription factors. However, under low-dose pinometostat treatment, where the expression of HOXA9 and MEIS1 is not affected, thousands of genes display differential expression, along with a disproportionate depletion of H3K79me2. Among the downregulated genes, the FLT3-ITD signaling axis is a major pathway that is sensitive to low-dose pinometostat treatment. Loss-of-FLT3-function recapitulates the cytotoxicity and gene expression consequences of low-dose pinometostat, whereas overexpression of constitutively active STAT5A, a target of FLT3-ITD-signalling, largely rescues these defects. These results suggest that co-occurring FLT3-ITD mutations sensitize MLL-r leukemias to DOT1L inhibition. Mechanistically, H3K79me2 depletion changes the distribution of other histone modifications, including genome-wide increases in H3K4me3 and a global reduction in H3K27me3. DOT1L inhibition downregulates the expression of EZH2 and EED components of the PRC2 complex, likely accounting for global reductions in H3K27me3 and cell proliferation defects. Overall, this is a comprehensive study that reveals interesting H3K79me dynamics and how it contributes to the leukemic transcription program. This study reveals alternative mechanisms to the canonical DOT1L-H3K79me2-HOXA9/MEIS1 axis in MLL-r leukemia, providing novel insights into epigenetic regulation and transcriptional controls as well as potentially new approaches for leukemia therapy. Addressing the following concerns would improve the manuscript:

1. The finding that the FLT3-ITD-signalling pathway is more sensitive to DOT1L inhibition than the canonical MLL-fusion targets (HOXA9 and MEIS1) is very interesting. However, all experiments are carried out in MLL-r leukemia cell lines, which somewhat limits the impact of the study. It would greatly enhance the impact of this study if the authors could look into leukemia cells beyond MLL-rearrangement. Specifically, does low-dose pinomeostat suppress the FLT3-STAT5A pathway and show therapeutic efficacy in FLT3-ITD+ leukemia w/o the MLL-rearrangement? Of note, FLT3-ITD occurs in 25% AML that have MLL PTD. It would be interesting to determine the effect of low-dose pinomeostat in some MLL-PTD cell lines.

2. The RNA-seq data have three replicates each, which is good. However, most of the ChIP-seq data appear to have no biological replicates. As many of the conclusions are largely based upon ChIP-seq results, biological repeats are needed for all ChIP-seq data.

3. All the experiments are performed using a single DOT1L inhibitor, pinometostat. To exclude potential off-target effects, an independent DOT1L inhibitor should be tested in some key experiments.

4. It is interesting that there is an increase of H3K4me after DOT1L inhibition. The authors propose it is due to Pol lI stalling, which leads to accumulation of H3K4me3. If this is the case, then accumulation of H3K4me3 might be more significant at highly expressed genes. However, the increase of H3K4me3 seems to be high regardless of absolute expression levels. There is also significant increase of H3K4me3 at gene bodies in the 'MLL-AF4 spread' genes. These need to be taken into consideration. Are the genes that show increase of H3K4me3 responsive to MLL inhibitor treatment?

---

## [Author Response]

1. The finding that the FLT3-ITD-signalling pathway is more sensitive to DOT1L inhibition than the canonical MLL-fusion targets (HOXA9 and MEIS1) is very interesting. However, all experiments are carried out in MLL-r leukemia cell lines, which somewhat limits the impact of the study. It would greatly enhance the impact of this study if the authors could look into leukemia cells beyond MLL-rearrangement. Specifically, does low-dose pinomeostat suppress the FLT3-STAT5A pathway and show therapeutic efficacy in FLT3-ITD+ leukemia w/o the MLL-rearrangement? Of note, FLT3-ITD occurs in 25% AML that have MLL PTD. It would be interesting to determine the effect of low-dose pinomeostat in some MLL-PTD cell lines.

We would like to thank the reviewers for these excellent suggestions—we initially tried to source a number of cell lines that had FLT3-ITD, MLL-PTD or both genetic lesions combined through existing repositories (ATCC and DSMZ) and requesting lines from other investigators. After an exhaustive effort that was no doubt limited by COVID-19 and ~$1200 in costs, the only lines we could obtain were PL-21(heterozygous FLT3-ITD with intact MLL1), and EOL-1 (MLL1-PTD with normal FLT3 status). Even these lines took > 2 months to arrive. Despite these challenges, the results are an impactful and exciting addition to this work – we present these new data in Figures 4H,I,J, and K.

The possibility that FLT3-ITD leukemias might respond to DOT1L inhibition is one of the more exciting implications from our first submission, and some preliminary testing of this concept in available cell lines certainly does boost the impact of our study. In new experiments, we find that PL-21 (FLT3^+/ITD^, MLL1^+/+^) cell viability was reduced after 10 µM pinometostat treatment for 9 days (Figure 4H), whereas at this dose we see no compromise to viability of K562 cells (an erythroleukemic cell line with a BCR-ABL translocation). Notably, in this FLT-ITD heterozygous line, there is a significant reduction of FLT3 and PIM1 expression upon treatment under conditions that begin to impact growth while canonical MLL-r drivers are below the limit of detection (Figure 4I). We also observe similar ablation of both H3K79me2 levels and STAT5A phosphorylation under these conditions (Figure 4J). These data are all consistent with our former speculation that FLT-ITD leukemias may be susceptible to DOT1L inhibition via the transcriptional down-regulation of FLT3, and consequent reduction in signaling activity. Moreover, this heterozygous FLT3-ITD leukemia line represents a conservative case, there is reason to anticipate a homozygous FLT3-ITD leukemia would be even more sensitive, as it is in the MLL-r backgrounds (Figure 4A). Although our studies are restricted to one cell line, these new data make a much more compelling case that this concept should be explored clinically in FLT3-ITD mutant AML (clinical frequency ~ 30-40%).

Unfortunately, we were unable to obtain an MLL1-PTD line that also bears a FLT3-ITD mutation. However, we were able to perform experiments that extend our MLL-AF4/AF9 translocation line experiments into the domain of MLL-PTD mutants that lack FLT3 mutations, which are more common in clinical presentation of acute leukemia (5-10%) than the MLL-r lesions we previously deployed. In brief, we observe highly similar proliferation defects in EOL-1 (FLT3^+/+^, MLL1^+/PTD^) as compared to MLL-r cell lines (Figure 4A and H). In EOL-1 cells, 100 nM pinometostat treatment for 7 days sharply reduced FLT3 expression with no observable effect on HOXA9, MEIS1 or PIM1 expression (Figure 4K). However, pinometostat treatment at a much higher dose (10 µM) reduced both HOXA9 and PIM1 expression (Figure 4K). These results are consistent with previous observations (Kuhn et al. Haematologica. 2015) where lower pinometostat doses are able to reduce the survival of EOL-1 rat xenografts without affecting HOXA9 expression. However, we observe that FLT3 expression is reduced at this lower concentration, perhaps accounting for the observed reduction in proliferation. Although we did not observe reduced phosphorylation of STAT5A upon 100 nM pinometostat treatment in EOL-1 cells (Figure 4J), this is not surprising as previous studies have found that WT FLT3 does not typically activate STAT5A (Choudhary et al., Int. J. Hematol. 2005). WT FLT3 is upregulated by MLL-fusions and is able to activate other pathways involved in cell growth and proliferation such as PI3K/AKT, RAS and AP-1 (Armstrong et al., Nat. Gent. 2002, Choudhary et al. Int. J. Hematol. 2005, Cauchy et al., Cell Rep. 2015). One or more of the aforementioned alternative growth signaling pathways may be essential for EOL-1 and MLL-PTD leukemia cell survival.

These new data strengthen the case that the sensitivity of the FLT3 locus to DOT1L inhibition also extends beyond MLL-rearranged leukemia to not only MLL-PTD but, leukemias bearing FLT3-ITD and intact MLL1.

2. The RNA-seq data have three replicates each, which is good. However, most of the ChIP-seq data appear to have no biological replicates. As many of the conclusions are largely based upon ChIP-seq results, biological repeats are needed for all ChIP-seq data.

In our initial submission we had performed one replicate of ICeChIP-seq to examine histone modification density upon pinometostat treatment at subsets of genes and specific gene loci. For conventional ChIP-seq this is perhaps insufficient for an analysis as central to the conclusions of our manuscript, however, among the advantages of ICeChIP-seq is the striking reproducibility of this method due to much lower experimental variance (Grzybowski et al., Mol. Cell. 2015, Shah et al., Mol. Cell. 2018). The cost of this reproducibility is the requirement for much greater sequencing depth of input, typically ~400-500 M reads per replicate, and to restrict the analysis to mono-nucleosomes, this must be paired-end data. This becomes extremely expensive per replicate ($3800). Nevertheless, we appreciate the broader point about reproducibility, and to demonstrate it unambiguously in this work, we present independent triplicate ICeChIP-seq measurements for the most important comparisons of H3K79me2 ICeChIP-seq in MV4;11 cells treated for 7 days with 100 nM pinometostat versus DMSO treated controls. Our two additional replicates of H3K79me2 ICeChIP-seq with 100 nM pinometostat treatment for 7 days very closely recapitulate all the effects, magnitudes, and features of the initial replicate to the point of being indistinguishable (Figure S3D (replicates 2 and 3), compare to replicate 1 in Figure 3D). These data are further supported by an independent experiment with only 4 days of treatment (Figures 3D, 3E S3A) that display the progressive H3K79me2 loss and mirrored the enrichment of H3K79me2 at MLL-AF4 targets and downregulated genes. For these comparisons we have also have added replicates of DMSO-treated cells (Figure S3D). The superb reproducibility of ICeChIP is demonstrated by scatterplot comparisons of the DMSO-treated H3K79me2 HMD from 0 to +2000 bp from the TSS at all genes expressed. We observe a strong correlation of H3K79me2 density in the gene bodies not only among replicate IPs performed side-by-side (Figure S3C (left)), but also on separate occasions months apart by a different person (Figure S3C (right)).

ENCODE and modENCODE have advocated duplicates of ChIP-seq data, coupled to metrics of irreproducibility amongst replicates (Landt et al. Genome Res. 2012). Compared to their example of a “highly reproducible” ChIP experiment, our HMD data from ICeChIP is far more correlated, even as compared to a log-log scale presentation (Figure S3C).

As they are an ancillary to the major points of the paper, we had trouble justifying $15,200 for additional H3K4me3 and H3K27me3 ICeChIP measurements under these conditions, but rather confirmed our findings with two additional independent ICeChIP-qPCR experiments for each modification at a number of key loci (Figures S6H and S7D). In support, we also now include a previously obtained independent ICeChIP-seq experiment for H3K4me3 after 100 nM pinometostat treatment for only 4 days which also displays increases in methylation even in this shorter timeframe (Figure S7D, compare to Figure 7C). Our two experimental replicates of H3K27me3 ICeChIP-qPCR ± 100 nM pinometostat display reductions in methylation at genes both up and downregulated by pinometostat, confirming our initial observations of global reductions in H3K27me3 by ICeChIP-seq and Western blot (Figure S6H, compare to 6B and 6C). Likewise, the two additional H3K4me3 ICeChIP experiments show increases in this modification upon pinometostat treatment near transcription start sites by qPCR (Figure S7E), consistent with our initial ICeChIP-seq analysis (Figure 7C). And just as in our ICeChIP-seq analysis H3K4me3 is increased not only at promoters of MLL-AF4 targets but at non-MLL-AF4 target genes both up- and downregulated by pinometostat treatment suggesting a genome-wide increase in H3K4me3 not limited to genes activated by MLL-AF4.

3. All the experiments are performed using a single DOT1L inhibitor, pinometostat. To exclude potential off-target effects, an independent DOT1L inhibitor should be tested in some key experiments.

While pinometostat is extraordinarily selective (>37,000-fold over a large panel of histone methyltransferases; Daigle et al., Blood 2013), to exclude any potential off-target effects as causal in many of our critical experimental readouts, we employed SGC0946. SGC0946 is a distinct, but still highly selective DOT1L inhibitor (>100-fold selectivity for DOT1L over other histone methyltransferases; Yu et al., Nat. Comm. 2012). Like pinometostat, we found that low concentration SGC0946 treatment (50 nM) reduced MV4;11 viability without significantly affecting HOXA9 and MEIS1 expression (Figure S1A and B). Just as with pinometostat, low dose SGC0946 reduced H3K79me2 levels (Figure S4B), as well as FLT3-ITD expression and STAT5A signaling (Figure S4B). Importantly, overexpression of constitutively active STAT5A was able to rescue the proliferation of MV4;11 cells treated with SGC0946 to a similar extent (Figure S5E). Collectively, the strikingly similar effects of both pinometostat and SGC0946 on MLL-rearranged leukemia in these key assays powerfully argues that reductions in FLT3-ITD/STAT5A signaling and attendant viability losses are due to depletion of H3K79me2 and not some off-target effect of pinometostat.

4. It is interesting that there is an increase of H3K4me after DOT1L inhibition. The authors propose it is due to Pol lI stalling, which leads to accumulation of H3K4me3. If this is the case, then accumulation of H3K4me3 might be more significant at highly expressed genes. However, the increase of H3K4me3 seems to be high regardless of absolute expression levels. There is also significant increase of H3K4me3 at gene bodies in the 'MLL-AF4 spread' genes. These need to be taken into consideration. Are the genes that show increase of H3K4me3 responsive to MLL inhibitor treatment?

We suggested one possible explanation for the genome-wide increase in H3K4me3 after pinometostat treatment could be due to reduced transcriptional elongation, resulting in more dwell time of poll II complex-associated MLL1 and increased H3K4me3. If this was the case, we reasoned that H3K4me3 increases due to pinometostat treatment would be correlated with transcriptional activity and would be scale with transcriptional activity. However, in new analysis prompted by this question, we find that if we group genes by increasing FPKM as a readout of transcriptional activity, there is no correlation with increased H3K4me3 at more highly expressed genes (Figure S7I), suggesting that increased H3K4me3 is not due to pol II stalling. The reviewers rightfully pointed out that the “MLL-AF4 spreading” genes show an increase in H3K4me3 not just after at the transcription start site but, far into the gene body, consistent with previous observations of the downstream spreading of the MLL-fusion and MLL1 complexes at these genes. It is also possible that these modifications each inhibit the enzymatic activity of the other complex. If this were the case, we would expect to see increases in H3K4me3 that are inversely proportional to the reduction in H3K79me2. The anticorrelation between these two modifications after pinometostat treatment is very poor (Figure S7H, R^2^ = 0.0023), suggesting that the increase in H3K4me3 it is not merely due reduced inhibition from ablation of H3K79me2. We also note that MLL-AF4 targets downregulated by pinometostat (Figure S7H purple) and all MLL-AF4 targets (Figure S7H red) show greater reductions in H3K79me2 than mean, but as a whole only display modest increases in H3K4me3, slightly above the overall distribution increases. Another possibility is that the increase in H3K4me3 is the result of the loss of recruitment or activity of the H3K4me2 demethylase LSD1 (KDM1A), an alternative possibility that we have added to the Discussion section (p. 16). LSD1, a component of the MLL1 supercomplex (Nakamura et al., Mol. Cell. 2002), and an ELL complex that also contains components of pTEFb (Biswas et al. Proc. Natl. Acad. Sci. USA. 2011). LSD1 is necessary for the leukemogenic expression program in murine MLL-AF9-transformed bone marrow cells (Harris et al. Cancer Cell. 2012). Knockdown or inhibition of this demethylase reduces the leukemogenic expression program, survival of MLL-AF9 leukemia cells and results in increases in H3K4me2/3 genome-wide (Harris et al., Cancer Cell. 2012, McGrath et al., Cancer Res. 2016). If H3K79me2 is necessary for full LSD1 activity or recruitment, reductions in H3K79me2 may result in an increase in H3K4me3. This is of course, speculative with the present data, and we are clear about the limits of our interpretation on this point in the discussion.

Related to the second question, we do not observe that the genes that show increases in H3K4me3 upon low-dose pinometostostat treatment are dependent on MLL1 function for their expression as treatment with the MLL1 inhibitor MI-503 has no effect on the expression of B2M, PIM2 and PBX3 (Figure 7F and Author response image 1), three genes that show moderate to large increases in H3K4me3 (Author response image 1).

**Author response image 1. respfig1:** (left) H3K4me3 HMD percent enrichment from ICeChIP-seq at the promoter regions of the indicated genes. (right) B2M expression fold-change ± 250 nM MI-503.